# Autophagy-mediated degradation of integumentary tapetum is critical for embryo pattern formation

Lin-lin Zhao [1], Ru Chen[1], Ziyu Bai [1], Junyi Liu [1], Yuhao Zhang [1], Yicheng Zhong [1], Meng-xiang Sun [1] & Peng Zhao [1,2] ✉

Autophagy modulates the degradation and recycling of intracellular materials and contributes to male gametophyte development and male fertility in plants. However, whether autophagy participates in seed development remains largely unknown. Here, we demonstrate that autophagy is crucial for timely programmed cell death (PCD) in the integumentary tapetum, the counterpart of anther tapetum, influencing embryo pattern formation and seed viability. Inhibition of autophagy resulted in delayed PCD of the integumentary tapetum and defects in embryo patterning. Cell-type-specific restoration of autophagic activities revealed that the integumentary tapetum plays a non-autonomous role in embryo patterning. Furthermore, high-throughput, comprehensive lipidomic analyzes uncovered an unexpected seed-developmental-stage-dependent role of autophagy in seed lipid metabolism: it contributes to triacylglycerol degradation before fertilization and to triacylglycerol biosynthesis after fertilization. This study highlights the critical role of autophagy in regulating timely integumentary tapetum PCD and reveals its significance in seed lipid metabolism and viability.

Autophagy is an evolutionarily conserved catabolic pathway in eukaryotes that degrades and recycles intracellular contents under normal and stress conditions[1,2]. There are three major types of autophagy: macro-autophagy, micro-autophagy, and chaperone-mediated autophagy[3]. Of these, macro-autophagy (hereafter referred to as autophagy) is the best studied and is characterized by the formation of a double-membrane autophagosome that encloses cytoplasmic components and facilitates their degradation and turnover[1]. Autophagy is tightly regulated by a set of conserved core *AUTOPHAGY-RELATED* (*ATG*) genes[4]. These core ATG proteins can be categorized into five functional groups during autophagosome formation: the ATG1–ATG13 complex, the ATG9 cycling system, the phosphatidylinositol 3-kinase (PI3K) complex, and ATG8 and ATG12 ubiquitin-like conjugation systems. The ATG8 conjugation system is composed of ATG3, ATG4, ATG7, and ATG8, while the ATG12 conjugation system is composed of ATG5, ATG7, ATG10, ATG12, and ATG16[4,5]. In plants,

autophagy is critical for adaptation to different abiotic and biotic environmental stresses[6,7]. In addition, recent evidence indicates that autophagy also plays crucial roles in normal development, including reproduction, in which it helps to shape pollen germination and anther development[8–12] as well as vegetative growth[13,14]. Despite its importance, little is known about the function of autophagy in early seed development[15].

In angiosperms, seeds develop from ovules following double fertilization and consist of three major parts: the embryo, the endosperm, and the seed coat. The embryo and endosperm are derived from two distinct fertilization events in which two sperm cells fertilize one egg cell and one central cell, which are surrounded by the seed coat[16–18]. The seed coat is a specialized tissue derived from maternal ovule integuments, which are usually composed of outer integument and inner integument layers. In some plant species, such as Arabidopsis (*Arabidopsis thaliana*) and tobacco (*Nicotiana tabacum*), the

[1]State Key Laboratory of Hybrid Rice, College of Life Sciences, Wuhan University, Wuhan, China. [2]Hubei Hongshan Laboratory, Wuhan, China. ✉e-mail: pzhao2000@whu.edu.cn

innermost layer of the inner integument differentiates into a specific tissue known as the integumentary tapetum or endothelium, which is the counterpart of the anther tapetum[19,20]. The anther tapetum provides protection, nutrients, enzymes, and precursors to developing pollen grains; importantly, this tissue degenerates through programmed cell death (PCD)[21–23] in a process that is necessary for pollen wall development and male fertility[24]. Early studies suggest that the integumentary tapetum serves multiple functions, including facilitating nutrient transport to the embryo sac and safeguarding the embryo sac to ensure the development of endosperm and embryo[19,25,26]. Similar to the anther tapetum, the integumentary tapetum also undergoes degeneration through PCD after fertilization, which initiates at the pre-embryo stage[25,27]. However, in comparison to the extensively studied anther tapetum in relation to pollen development and male fertility, the role of the integumentary tapetum in seed development and molecular mechanisms regulating timely integumentary tapetum degradation are not well understood.

Our previous comprehensive expression profile analysis of *ATG* genes revealed their relatively high expression level and temporal dynamics during tobacco seed development[28], suggesting an important role for autophagy in this process. Here, we provide evidence that autophagy is critical to the timely degradation of the integumentary tapetum and proper embryo pattern formation during seed development. Suppression of autophagy by mutation of the autophagy-specific ATG5 or ATG7, two essential components in the ubiquitin-like conjugation systems necessary for autophagosome formation and maturation[29], resulted in delayed PCD of integumentary tapetum and culminated in embryo death. Comparative lipidomic analyzes revealed that autophagy plays a dual role in lipid metabolism during seed development: contributing to lipid degradation before fertilization and supporting lipid biosynthesis after fertilization. Our findings demonstrate that autophagy is an essential mechanism for controlling timely PCD of the integumentary tapetum, which is critical for correct embryonic pattern formation and early seed development.

## Results

### Autophagy is active in the integumentary tapetum

Autophagy is an evolutionarily conserved intracellular degradation mechanism that is controlled by a set of *ATG* genes[4]. Our previous genome-wide analysis implicated most *ATG*s during seed formation based on their temporal expression patterns[28], suggesting the potential importance of autophagy in seed development. To explore this possibility, we undertook three different approaches to investigate autophagic activities during seed development. First, we monitored the expression of two autophagy-specific *ATG*s, *ATG5,* and *ATG7*, which are responsible for autophagosome formation, by visualizing the fluorescence of green fluorescent protein (GFP) fused to histone H2B (H2B, to concentrate GFP in nuclei) in *proATG5:H2B-GFP* and *proATG7:H2B-GFP* transgenic plants in tobacco. This showed that *ATG5* is widely expressed in all layers of the integuments, including the innermost integumentary tapetum (Fig. 1a). We also detected GFP fluorescence in zygotes or early embryos at different developmental stages (Supplementary Fig. 1). *ATG7* displayed a similar expression pattern to *ATG5*, as we observed GFP fluorescence in all layers of integuments including the integumentary tapetum, and early embryos during seed development (Fig. 1b and Supplementary Fig. 1). In addition, both ATG5 and ATG7 proteins are located in the cytoplasm of integumentary tapetum cells (Supplementary Fig. 2a, b). Second, we visualized autophagosomes through ATG8 immunofluorescence[9,30], as well as by inspecting GFP fluorescence from the *proTPE8:GFP-ATG8* marker line, in which expression of the *GFP-ATG8* cassette is driven from the integumentary tapetum-specific promoter *NtTPE8*[25] (Fig. 1c, d and Supplementary Fig. 3). We detected clear, distinct puncta in the integumentary tapetum by both methods. Transmission electron microscopy (TEM) analysis further confirmed the presence of active

autophagy in the integumentary tapetum during seed development. Indeed, we observed double-membrane autophagosomes in the cytoplasm and autophagic bodies in the integumentary tapetum (Fig. 1e). These findings indicate that autophagy is active during seed development, notably in the integumentary tapetum.

### Autophagy is essential for seed development

To investigate the role of autophagy in seed development, we generated *atg5* and *atg7* mutants using the clustered regularly interspaced short palindromic repeat (CRISPR)–CRISPR-associated nuclease 9 (Cas9)-mediated genome editing system as previously reported[9]. We obtained three independent mutant alleles each of *ATG5* and *ATG7* for phenotypic analysis (Fig. 2a, b). Western blot analysis revealed that the levels of ATG5 and ATG7 proteins were markedly reduced in their respective mutant lines (Fig. 2c). The autophagic activities of *atg5-2* and *atg7-1* mutant plants were first assessed by ATG8 immunofluorescence assay (Fig. 2d, e and Supplementary Fig. 4) and with the *proTPE8:GFP-ATG8* marker line (Fig. 2f, g). We detected punctate autophagosomes in the integumentary tapetum of wild-type (WT) plants, but hardly the *atg5-2* and *atg7-1* mutant plants (Fig. 2d–g and Supplementary Fig. 4). We subsequently assessed the extent of ATG8 conjugated with phosphatidylethanolamine (PE), known to be crucial for autophagosome formation, to further determine autophagic activities in *atg5-2* and *atg7-1* mutants. Consistent with the results from the ATG8 immunofluorescence assay, we observed a notable decrease in the levels of lipidated ATG8 in both *atg5-2* and *atg7-1* mutants compared to the WT plants (Fig. 2h). These results indicate that mutations in *ATG5* or *ATG7* impair autophagosome formation and lead to reduced autophagic activities. Moreover, *atg5* and *atg7* mutants displayed defects in seed development, as we noticed about 30% aborted seeds in all *atg5* and *atg7* mutant lines, which was significantly higher than in WT plants (Fig. 2i, j). We fully rescued the seed-setting defect of *atg5* plants by introducing a *proATG5:ATG5-GFP* transgene (Fig. 2i, j), demonstrating the essential role of autophagy activity in seed development and the functionality of ATG5-GFP in regulating autophagy in the seed.

### Autophagy contributes to endothelium and embryo development

Seeds from angiosperms are composed of three major compartments—the seed coat, endosperm, and embryo—whose coordinated development is crucial for seed formation[31]. To investigate the involvement of autophagy in seed development, we carefully analyzed the development of these three compartments. Before fertilization, we observed no obvious differences between WT and *atg5* or *atg7* ovules. After fertilization, however, seeds from the *atg5* and *atg7* mutants displayed morphological abnormalities as early as 4 days after pollination (DAP), which prompted us to examine transverse sections of the ovule and seed at different developmental stages (Fig. 3a, b). We observed clear developmental defects in the integumentary tapetum and the embryo of *atg5* and *atg7* mutants (Fig. 3 and Supplementary Figs. 5 and 6). In WT seeds, the integumentary tapetum expanded from 0 to 4 DAP, and then gradually degraded. In contrast to the WT, the integumentary tapetum of *atg5* and *atg7* mutant seeds became extremely expanded. The integumentary tapetum of *atg5* and *atg7* mutant seeds continued to expand after 4 DAP and became extremely thick compared to that in WT seeds at 6 DAP (Fig. 3a, b and Supplementary Fig. 6).

To further explore the integumentary tapetum phenotype of *atg* mutant seeds, we used a modified pseudo-Schiff-propidium iodide (mPS-PI) staining technique (Fig. 3c, Supplementary Fig. 6a, and Supplementary Movies 1 and 2) to visualize integumentary tapetum cells in whole-mount WT and *atg* ovules and seeds[32]. The architecture of the integumentary tapetum was indistinguishable between the WT and *atg5-2* mutants before fertilization. However, we observed defects in

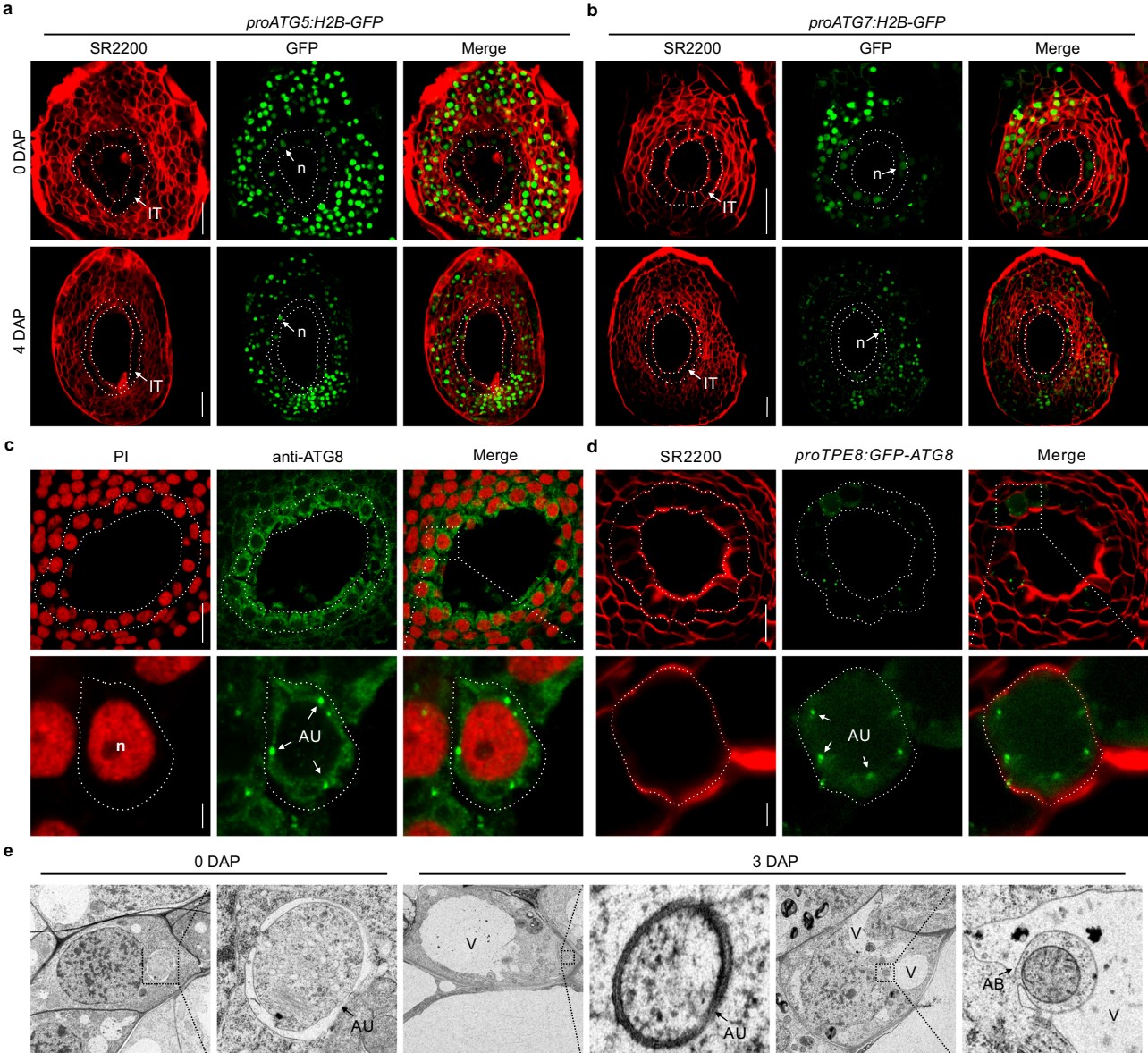

**Fig. 1 | Autophagy is active in the integumentary tapetum during seed development. a, b** Analysis of *proATG5:H2B-GFP* (**a**) and *proATG7:H2B-GFP* (**b**) reporter lines revealed that both *ATG5* and *ATG7* are expressed in the integumentary tapetum. The cell wall was stained by SCRI Renaissance 2200 (SR2200), and the integumentary tapetum was delineated with dot lines. DAP, days after pollination; IT integumentary tapetum, n nucleus. Scale bars: 50 μm. **c** Autophagosomes in the integumentary tapetum were labeled by ATG8 immunofluorescence. The nuclei were co-stained with propidium iodide (PI). The lower row shows the magnified integumentary tapetal cells, which are outlined with the dot lines. Scale bars: 20 μm (upper row); 2.5 μm (lower row). **d** GFP-ATG8 was expressed in the integumentary tapetum driven by the tapetum-specific promoter *NtTPE8*. The integumentary

tapetum and magnified integumentary tapetal cell were delineated with dot lines. Scale bars: 20 μm (upper row); 2.5 μm (lower row). **e** Transmission electron microscopy assays show typical autophagosomes and autophagic bodies in the integumentary tapetum. AU autophagosomes, AB autophagic bodies, V vacuole. Scale bars: 2 μm (left); 200 nm (right). Observation of GFP in *proATG5:H2B-GFP* (**a**) and *proATG7:H2B-GFP* (**b**), as well as the visualization of autophagosomes in the integumentary tapetum through ATG8 immunofluorescence (**c**), and the observation of GFP from the *proTPE8:GFP-ATG8* (**d**) were repeated three times with similar results. The transmission electron microscopy experiment was performed two times with similar results.

integumentary tapetum degeneration in *atg5-2* mutant seeds (Fig. 3c, d). This degradation defect of integumentary tapetum in the *atg5-2* mutant was fully rescued by introducing the *proATG5:ATG5-GFP* transgene into the mutant background (Supplementary Fig. 5). Importantly, a similar phenotype of integumentary tapetum development was also observed in the *atg7-1* mutant (Supplementary Fig. 6a–c). We conclude that a defect in autophagy suppresses the degradation of the integumentary tapetum during seed development.

We also detected defects in embryo pattern formation in the *atg5* and *atg7* mutants. Abnormal seeds were recognizable among *atg*

mutant seeds compared to WT seeds at 6 DAP; we, therefore, subjected 6-DAP seeds to mPS-PI staining. In contrast to the WT, a significantly higher portion of embryos from the *atg5-2* (39%) and *atg7-1* (40%) mutant seeds exhibited abnormalities in the development of both apical and basal cell lineages (Fig. 3e and Supplementary Fig. 6d). We grouped these embryos from *atg5* mutant into four major classes according to their embryonic cell division pattern (Fig. 3e). Compared to WT embryos, 18% of the embryos showed an abnormal cell division pattern in the apical cell lineage, while 13% of the embryos lacked a typical suspensor in *atg5-2* mutant.

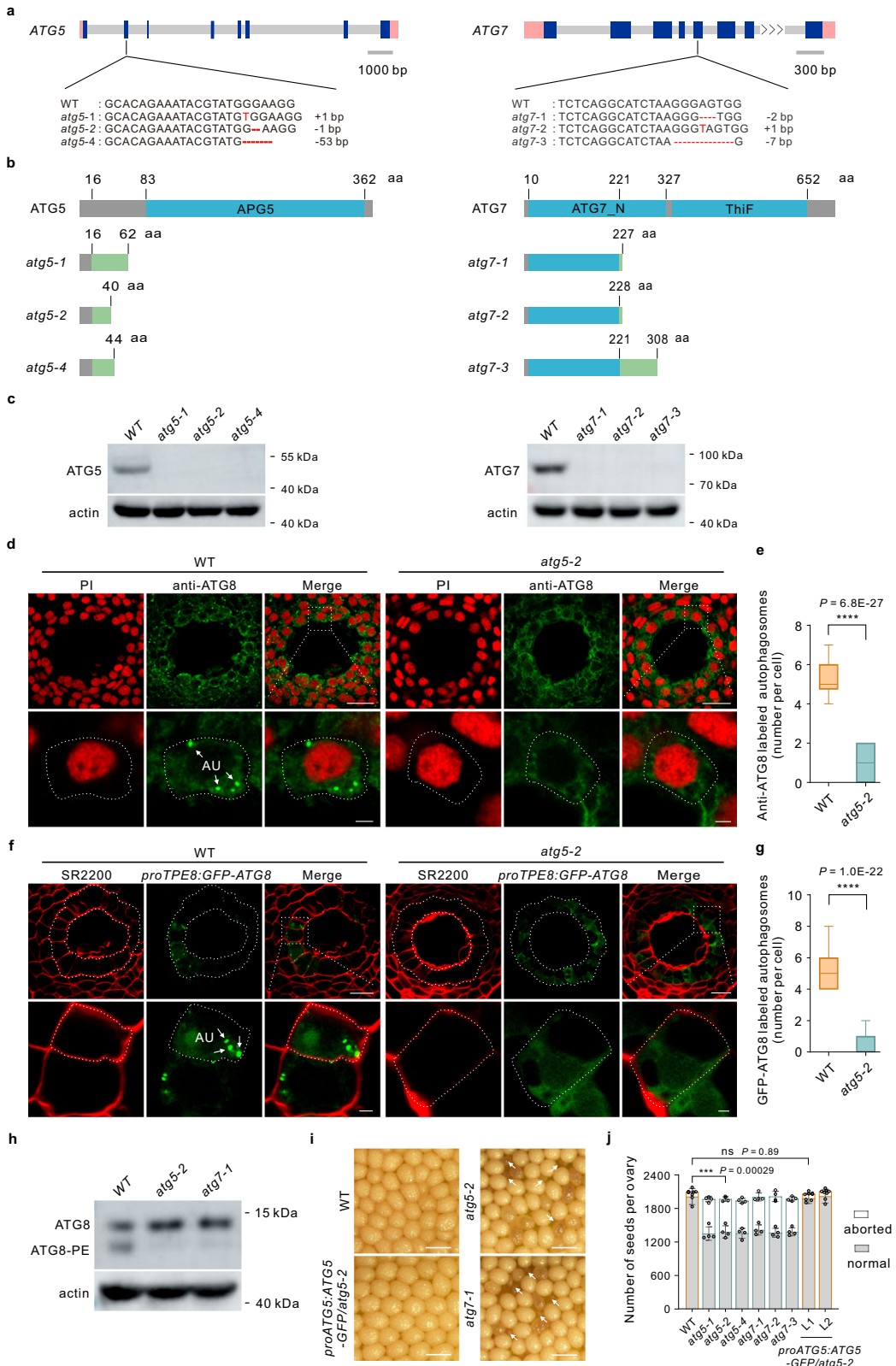

Similar embryo developmental defects were also observed in the *atg7-1* mutant, with 19% of the embryos showing abnormal development of apical cell lineage and 12% of the embryos lacking a typical suspensor (Supplementary Fig. 6d). Taken together, these results indicate that autophagy is also critical for the establishment of apical–basal embryo polarity.

## Autophagy is crucial for timely integument tapetal PCD onset

Seed development involves a tightly controlled and stage-dependent PCD of the seed coat, endosperm, and suspensor, resulting in their degradation[33–35]. To investigate the role of autophagy in the PCD of integumentary tapetal cells, we utilized terminal deoxynucleotidyl-transferase (TdT)-mediated dUTP nick-end labeling (TUNEL)[36] and

**Fig. 2 | Mutation in *ATGs* blocks autophagosome formation and leads to seed abortion. a** *atg5* and *atg7* Mutants were generated by CRISPR/Cas9 system. Schematic diagrams of *ATG5* and *ATG7* genomic sequences. Blue box, exon; gray line, intron; and pink box, untranslated region. The inserted or deleted nucleotides are indicated under the diagram. **b** Schematic diagrams showing the domains within the ATG5 and ATG7 proteins, as well as the mutant versions created by CRISPR/Cas9-edited mutagenesis. Missense sequences resulting from frameshift or fragment deletion are depicted as green boxes. **c** Protein levels of ATG5 and ATG7 in the wild type (WT) and their corresponding *atg* mutants were evaluated. β-Actin was used as the loading control. The immunoblot assay was repeated three times with similar results. **d, f** Representative images showing ATG8 immunofluorescence-labeled autophagosomes (**d**) and GFP-ATG8-labeled autophagosomes (**f**) in the integumentary tapetum from the WT and *atg5-2* mutant. The lower rows show a magnified integumentary tapetal cell, which is outlined with the dot lines. PI propidium iodide, SR2200 SCRI Renaissance 2200. Arrows indicate autophagosomes.

Scale bars: 20 μm (upper row) and 2.5 μm (lower row) in (**d**); 20 μm (upper row) and 2.5 μm (lower row) in (**f**). **e, g** The number of ATG8 immunofluorescence-labeled autophagosomes (**e**) and GFP-ATG8-labeled autophagosomes (**g**) in the integumentary tapetum are presented in box and whisker plots. Centerline, median values; top and bottom of boxes, 75th and 25th percentiles; top and bottom lines, maximum and minimum values. Data are from three independent experiments, each experiment with ten samples ($n = 30$). **h** The level of ATG8–PE adduct decreased in *atg5-2* and *atg7-1* mutants. β-actin serves as the loading control. ATG8–PE assay was repeated three times with similar results. **i, j** Mutation in *ATG5* and *ATG7* results in seed abortion. **i** Representative images showing seed set from the WT, *atg5-2* and *atg7-1* mutants, and *proATG5:ATG5-GFP* complementation lines. Arrows indicate aborted seeds. Scale bars: 1 mm. **j** The number of normal and aborted seeds in the WT, *atg5* and *atg7* mutants, and the complementation lines. Values are means ± SD ($n = 4$ independent ovaries), (two-tailed Student's *t* test, ns no significant difference, $P > 0.05$; ***$P < 0.001$; ****$P < 0.0001$).

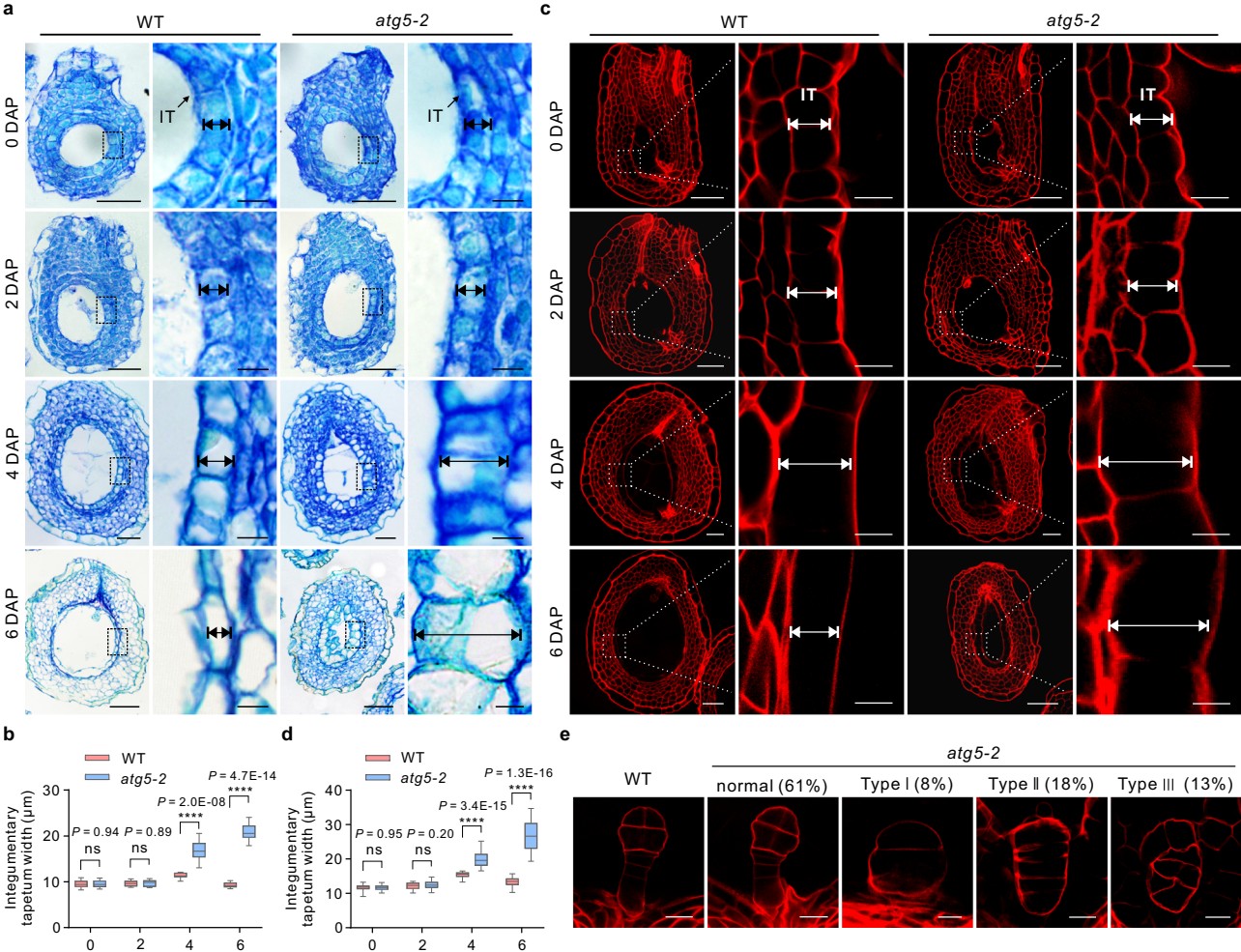

**Fig. 3 | Deficiency of autophagy leads to defects in integumentary tapetum degradation and embryo pattern formation. a, c** Paraffin sections combined with toluidine blue staining (**a**) and modified pseudo-Schiff-propidium iodide (mPS-PI) staining (**c**) were to show the ovules and seeds at different developmental stages from the WT and *atg5-2* mutant. The right images show the magnified area outlined by the dotted lines. Two-way arrows indicate integumentary tapetum for width quantification. IT integumentary tapetum, DAP days after pollination. Scale bars: 50 μm (left) and 10 μm (right) for 0-DAP ovules, 2-DAP seeds, and 4-DAP seeds; 100 μm (left) and 10 μm (right) for 6-DAP seeds. **b, d** The width of integumentary tapetum of the ovules and seeds from the WT and *atg5-2* mutant, which were visualized through paraffin section combined with toluidine blue staining (**b**) and

mPS-PI staining (**d**). The data of integumentary tapetum width are displayed in box and whisker plots. The center line represents the median, while the bottom and top edges of each box represent the 25th and 75th percentiles. The whiskers indicate the minimum and maximum values. (**b** $n = 18$ for 0-DAP ovules and 2-DAP seeds, $n = 15$ for both 4-DAP and 6-DAP seeds, **d** $n = 30$ ovules or seeds). **e** Embryos with abnormal cell division patterns were observed in the *atg5-2* mutant. The embryos of *atg5-2* mutant could be classified into four major different types according to embryonic cell division pattern ($n = 100$ independent embryos). Scale bars: 20 μm. A two-tailed Student's *t* test was performed for statistically significant difference analysis (ns, no significant difference, $P > 0.05$; ****$P < 0.0001$).

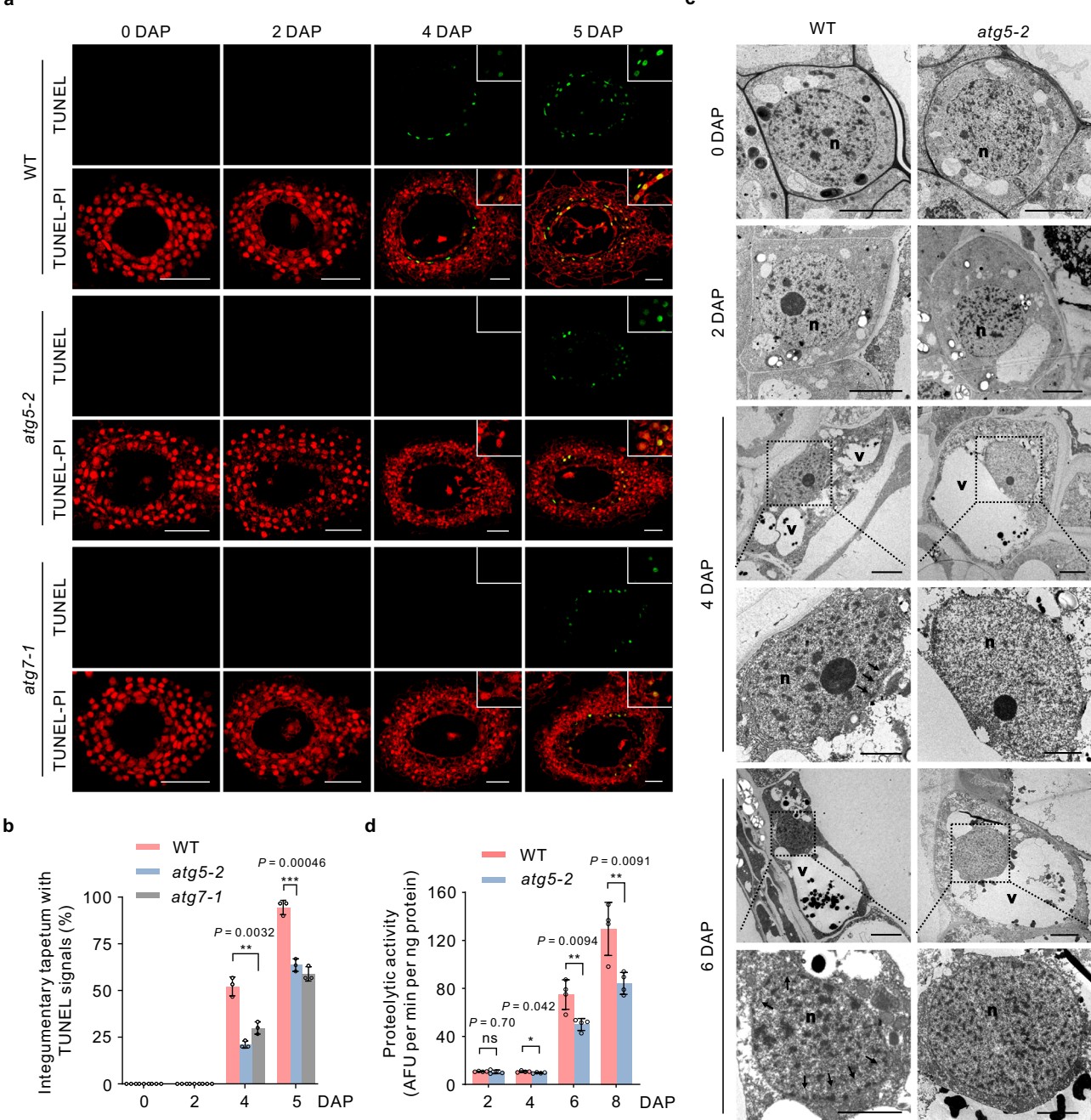

**Fig. 4 | Deficiency of autophagy results in the delayed programmed cell death of integumentary tapetum. a** TUNEL assay showing the DNA fragmentation in the integumentary tapetum of the WT, *atg5-2* and *atg7-1* mutants. PI propidium iodide, TUNEL terminal deoxynucleotidyltransferase (TdT)-mediated dUTP nick-end labeling, DAP days after pollination. Scale bars: 50 μm. **b** The frequencies of integumentary tapetum with TUNEL signals in the WT, *atg5-2* and *atg7-1* mutants. Data are means ± SD from three independent experiments, each experiment with 30 samples (*n* = 90). **c** Morphology of the integumentary tapetal cells in ovules and seeds from the WT and *atg5-2* mutant revealed by transmission electron microscopy. Arrows indicate a disintegrating nuclear envelope. n nucleus, v vacuole. Scale bars: 2.5 μm for the enlarged view of 4-DAP and 6-DAP seeds, and 5 μm for the others. Transmission electron microscopy assay was performed three times with similar results. **d** Proteolytic activities of vacuolar processing enzyme in the ovules and seeds at three developmental stages in both WT and *atg5-2* plants. Data represent the mean ± SD from four independent experiments (*n* = 4), (two-tailed Student's *t* test; ns, no significant difference, *P* > 0.05; *P < 0.05, **P < 0.01, ***P < 0.001).

TEM analyzes. We detected no TUNEL signals in the WT integumentary tapetum at 0 or 2 DAP (Fig. 4a, b). However, TUNEL signals in the integumentary tapetum (52.2% seeds; 47 out of 90 seeds) began to appear at 4 DAP, and nearly all integumentary tapetal cells (94.4% seeds; 85 out of 90 seeds) showed signals at 5 DAP in the WT (Fig. 4a, b), indicating that PCD in the integumentary tapetum is precisely regulated. Compared to the WT, PCD in the integumentary

tapetum of the *atg5-2* mutant was delayed: only 21.1% seeds (19 out of 90 seeds) of *atg5-2* mutant showed initiation of DNA fragmentation in the integumentary tapetum as early as 4 DAP, and we detected significantly fewer TUNEL signals (63.3%; 57 out of 90) in the integumentary tapetum at 5 DAP (Fig. 4a, b). Similar delayed PCD was also detected in the integumentary tapetum of *atg7-1* mutants (Fig. 4a, b). Furthermore, TEM analysis revealed that the nuclei of WT

integumentary tapetal cells became lobed and their nuclear envelope appeared ruptured at 4 DAP before being progressively disassembled (Fig. 4c); this was accompanied by the typical cytoplasmic vacuolization seen during vacuolar PCD in other plant cells[37]. By contrast, the nuclear membrane of expanded integumentary tapetal cells remained intact in the *atg5-2* mutant even at 6 DAP (Fig. 4c). Vacuolar processing enzyme (VPE), which possesses caspase 1-like activity, is a critical protease in the execution of plant PCD[38–40]. To determine the relationship between caspase 1-like protease VPE and autophagy in the PCD of the integumentary tapetum, we compared the proteolytic activities of VPE during WT and *atg5-2* seed development. Consistent with the delayed PCD in autophagy-deficient mutants, we observed a significant decrease in VPE proteolytic activities in the 4-DAP seeds and subsequent stages in the *atg5-2* mutant, but not in the 2-DAP seed prior to PCD, as compared with the WT (Fig. 4d). This suggests that autophagy seems to lie upstream of caspase 1-like protease VPE in the signaling pathway of integumentary tapetum PCD. Overall, our results indicate that autophagy plays a crucial role in the timely initiation of PCD in the integumentary tapetum.

### Integumentary tapetum PCD promotes embryo development

Timely PCD in the anther tapetum, the innermost layer of the anther wall that surrounds and supplies nutrients to microspores, is critical for pollen development[23,24], whereas the role of integumentary tapetum in early embryo development has not been well understood. As described above, we observed developmental defects in both integumentary tapetum and embryo in the *atg5* mutants, providing us with a unique opportunity to investigate cell communication between integumentary tapetum and embryo. To explore the role of the integumentary tapetum, particularly its timely degradation, in embryo development, we drove *ATG5* expression with the integumentary tapetum-specific promoter *NtTPE8*[25] in the *atg5-2* mutant background. We detected the specific expression of *ATG5-GFP* in the integumentary tapetum, as evidenced by GFP fluorescence (Fig. 5a). We examined whether the specific expression of *ATG5* in the integumentary tapetum of the *atg5-2* mutant could functionally rescue the defects in timely integumentary tapetum PCD and early embryo development. Indeed, expression of *ATG5-GFP* in *atg5-2* integumentary tapetum partially rescued the defects in integumentary tapetum degradation seen in *atg5* mutant seeds (Fig. 5b, c), further confirming that autophagy is essential for the PCD of integumentary tapetum. Notably, the frequency of abnormal embryos with defects in cell division pattern also decreased significantly, from approximately 40% to 21%, in the *atg5-2 proTPE8:ATG5-GFP* complementation lines (Fig. 5d, e). Moreover, specific expression of *ATG5-GFP* in the *atg5-2* integumentary tapetum promoted embryo development and seed formation, as the seed-setting rate rose from 70.1% in the *atg5-2* mutant to 82.5% in *atg5-2 proTPE8:ATG5-GFP* complementation lines (Fig. 5f, g). These results indicate that autophagy-mediated timely PCD of the integumentary tapetum contributes to the correct formation of the embryo pattern in a non-cell-autonomous manner, and suggest a possible cell-to-cell communication between the integumentary tapetum and the embryo.

To investigate the molecular connections between autophagy in the integumentary tapetum and embryo pattern formation, we isolated embryos from WT plants, *atg5-2* mutant, and *atg5-2* mutant expressing *proTPE8:ATG5-GFP* for RNA sequencing. Overall, the transcripts from the same cell type, but across different biological replicates, exhibited high correlation ($r > 0.98$). Principal component analysis (PCA) and complete-linkage clustering showed that these nine transcriptomes were classified into three groups (Fig. 6a, b): WT, *atg5-2*, and *atg5-2 proTPE8:ATG5-GFP*. The intermediate positioning of the *atg5-2 proTPE8:ATG5-GFP* group between WT and *atg5-2* suggests that specific expression of *ATG5* in the integumentary tapetum of *atg5-2* could partially rescue the gene expression profile of *atg5-2* embryos. Consistent with the PCA results, a comparison of differentially

expressed genes (DEGs) between WT/*atg5-2* and *atg5-2 proTPE8:ATG5-GFP/atg5-2* revealed that the specific expression of *ATG5* in the integumentary tapetum of *atg5-2* mutant leads to a significant decrease in the number of DEGs in the *atg5-2* embryos (Fig. 6c–e and Supplementary Data 1). This reduction suggests that specific expression of *ATG5* in the integumentary tapetum can partially restore the aberrant embryonic gene expression profile observed in the *atg5-2* mutant. We then focused our analysis on the set of genes in embryos that are regulated by autophagy in the integumentary tapetum (Fig. 6f). Gene ontology (GO) analysis identified several well-known pathways associated with embryo pattern formation, including pattern specification and auxin-related pathways, which are regulated by autophagy in the integumentary tapetum (Fig. 6f). This finding further confirms the non-cell-autonomous role of autophagy in the integumentary tapetum during embryo development.

### Autophagy plays dual roles in seed lipid homeostasis

Previous reports have revealed a close relationship between autophagy and lipid metabolism in animals and plants[41,42]. In particular, lipids can serve as regulatory molecules and degradation substrates for autophagy[43,44], as well as play significant roles in seed development[45,46]. To investigate the potential role of autophagy in lipid metabolism during early seed development, we performed a comparative lipidomic analysis of ovules and seeds at 4 DAP from the WT and *atg5-2* mutant using ultra-high-performance liquid chromatography followed by tandem mass spectrometry (UPLC–MS/MS). Five biological replicates were performed for both the WT and *atg5-2* ovules, and 4-DAP seeds. Thus, a total of 20 independent samples were included for comparative lipidomic analysis. As a result, we detected 703–727 individual lipids in the lipidomic data of different samples, which could be grouped into 30 classes including triacylglycerol (TG), phosphatidylcholine (PC), phosphatidylinositol (PI), and PE (Supplementary Data 2). TG was the most abundant lipid in the ovules and seeds at 4 DAP, followed by PE, PC, and PI. PCA (Fig. 7a) uncovered a significant separation between ovules and seeds along the first principle component, explaining 50.1% of the standing variation, with the *atg5-2* and WT samples being separated along PC2 (explaining 17.9% of the variation), revealing the dynamics of lipid metabolism as a function of seed age and the effect of autophagy on lipid homeostasis during early seed development.

We focused on the differences in lipid composition between WT ovules and 4-DAP seeds to investigate lipid metabolism during early seed development (Fig. 7b and Supplementary Figs. 7 and 8). We detected a significant accumulation of numerous TG species (77 out of 225; Supplementary Data 2), the main components of stored lipids in plants, in seeds after fertilization (Fig. 7b and Supplementary Fig. 7a, b). We also observed the accumulation of 29 of 76 diacylglycerol (DG) species, the intermediates in TG biosynthesis, in the seed compared to unfertilized ovules (Fig. 7b and Supplementary Fig. 7a, b), consistent with the essential role of TG biosynthesis in seed development[45]. In addition, the two phospholipids PE and PI, which are involved in the formation of autophagosomes[43], accumulated in the seeds compared to the ovules (Fig. 7b and Supplementary Fig. 7a, b). However, when we compared the lipid compositions of *atg5-2* ovules and 4-DAP seeds (Supplementary Fig. 7c, d), we noted that a significant fraction of lipid species (163 out of 320; Supplementary Data 2) are differentially regulated compared to those in WT ovules and seeds (Supplementary Fig. 7e). Specifically, 29 out of 77 TG species were specifically more abundant following fertilization in WT plants, but not in the *atg5-2* mutant, confirming that autophagy has considerable influence on lipid metabolism during early seed development.

To explore this effect in more detail, we compared the lipid compositions of ovules and seeds from the WT and the *atg5-2* mutant (Fig. 7b and Supplementary Fig. 8). We detected hundreds of metabolites whose abundances differed between WT and *atg5-2* ovules

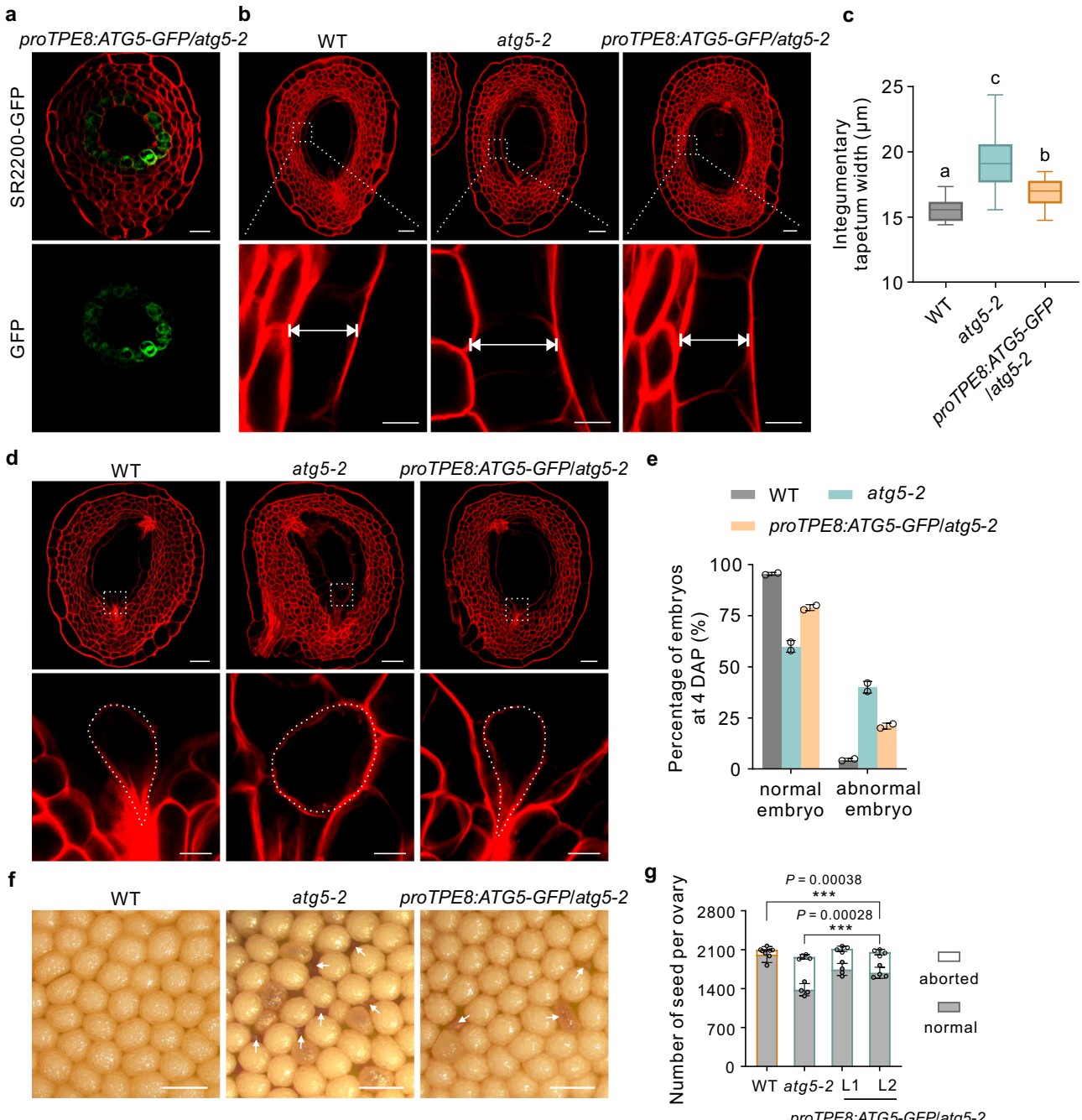

**Fig. 5 | Specific expression of *ATG5-GFP* in the integumentary tapetum of *atg5-2* mutant promotes embryo pattern formation. a** Specific expression *ATG5-GFP* in the *atg5-2* integumentary tapetum under the control of the integumentary tapetum-specific promoter *NtTPE8*. GFP GFP fluorescence, SR2200 SCRI Renaissance 2200. Scale bar: 20 μm. Observation of GFP in *proTPE8:ATG5-GFP* was performed three times with similar results. **b, d** mPS-PI staining showing representative integumentary tapetum (**b**) and embryos (**d**) in 4-DAP seeds from the WT, *atg5-2* mutant and *proTPE8:ATG5-GFP* complementation plants. Scale bars: 50 μm (upper row) and 10 μm (lower row). **c** Quantitative analysis of the integumentary tapetum width. Values displayed in box and whisker plots are from three independent experiments, with ten seeds in each experiment (*n* = 30). Centerline, median values; top and bottom of boxes, 75th and 25th percentiles; top and bottom lines, maximum and minimum values. Tukey–Kramer multiple comparison test was used to

determine the significant differences (one-way analysis of variance (ANOVA) between groups; *P* = 1.32E-15; *F* = 52.12). The labels **a**–**c** represent highly significant differences when compared at the 0.01 level. **e** Percentages of embryos identified in 4-DAP seeds from the WT, *atg5-2* mutant, and *proTPE8:ATG5-GFP* complementation plants. Data are the mean ± SD from two independent experiments, with 50 embryos analyzed in each experiment (*n* = 100). **f** Representative images showing seeds from the WT, *atg5-2* mutant, and *proTPE8:ATG5-GFP* complementation plants. Arrows indicate aborted seeds. Scale bars: 1 mm. **g** The number of normal and aborted seeds in the WT, *atg5-2* mutant and *proTPE8:ATG5-GFP* complementation plants. Gray boxes indicate the number of normal seeds, while the white boxes represent the number of aborted seeds. Values are means ± SD (*n* = 4 independent ovaries), (two-tailed Student's *t* test; ***P < 0.001).

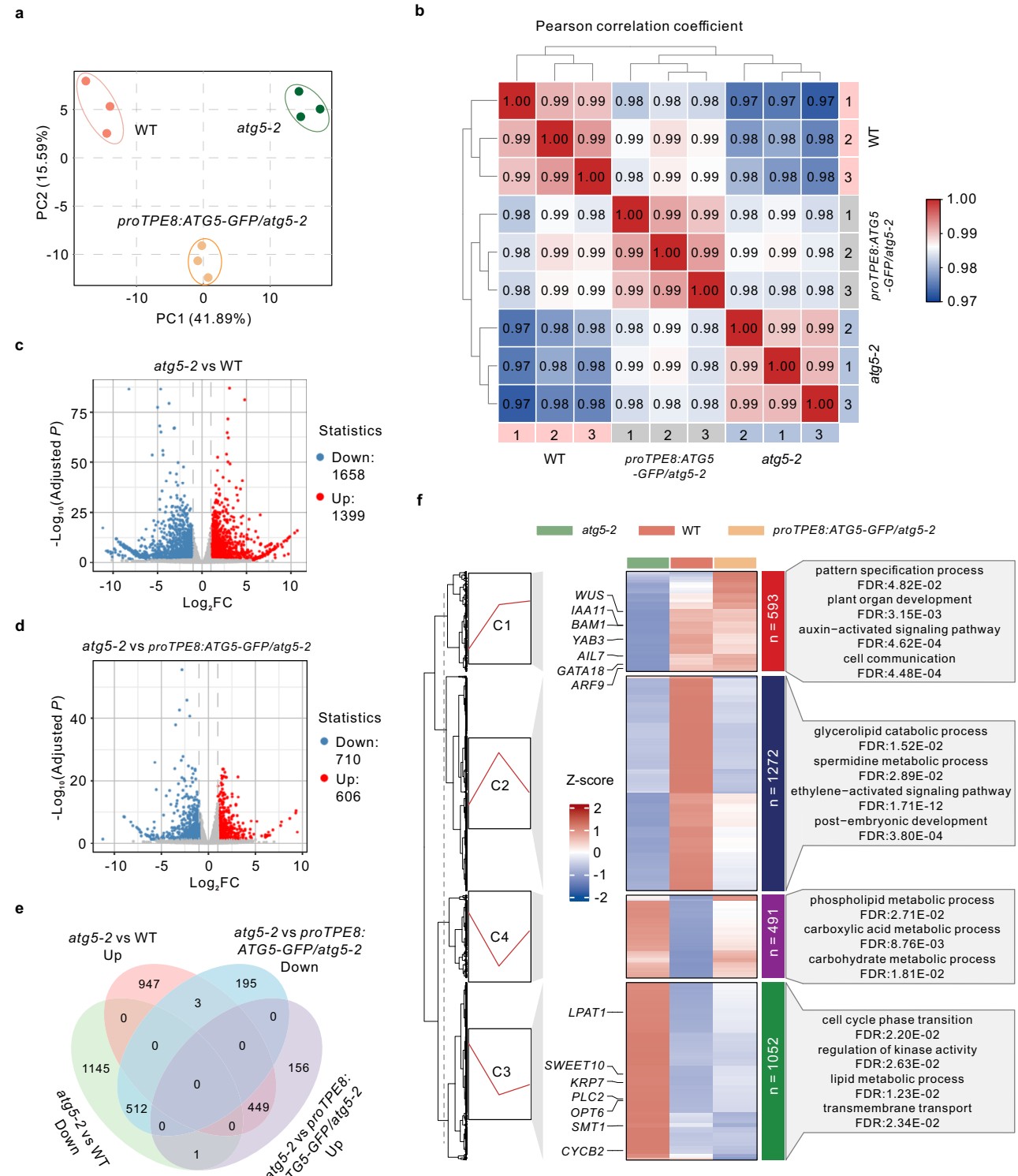

**Fig. 6 | Comparative transcriptomic analysis of 8-cell embryos from WT, *atg5-2* mutant, and *atg5-2* mutant complemented with integumentary tapetum-specific *ATG5* expression. a** Principal component analysis of transcriptomes of 8-cell embryos from the WT, *atg5-2* mutant, and *atg5-2* mutant expressing *proT-PE8:ATG5-GFP*. Three biological replicates for each genotype (*n* = 3). **b** Heat map showing Pearson's correlation across RNA-seq datasets. **c**, **d** Volcano plot showing differentially expressed genes between *atg5-2* and WT embryos (**c**) or between *atg5-2* and *atg5-2* mutant expressing *proTPE8:ATG5-GFP* embryos (**d**). Red dots indicate upregulated genes, while blue dots indicate downregulated genes. The

Benjamini–Hochberg method was used to adjust the two-sided *P* values for multiple comparisons. FC fold change, Adjusted *P*, Benjamini–Hochberg adjusted *P* value. **e** Comparisons of differential genes between *atg5-2*/WT and *atg5-2*/*atg5-2 proTPE8:ATG5-GFP*. **f** Differentially expressed genes between *atg5-2* and WT could be categorized into four distinct clusters. DEGs located in clusters I and III are regulated by ATG5 in the integumentary tapetum, whereas DEGs in clusters II and IV are not. Gene ontology enrichment analysis of genes in four different groups. FDR false-discovery rate.

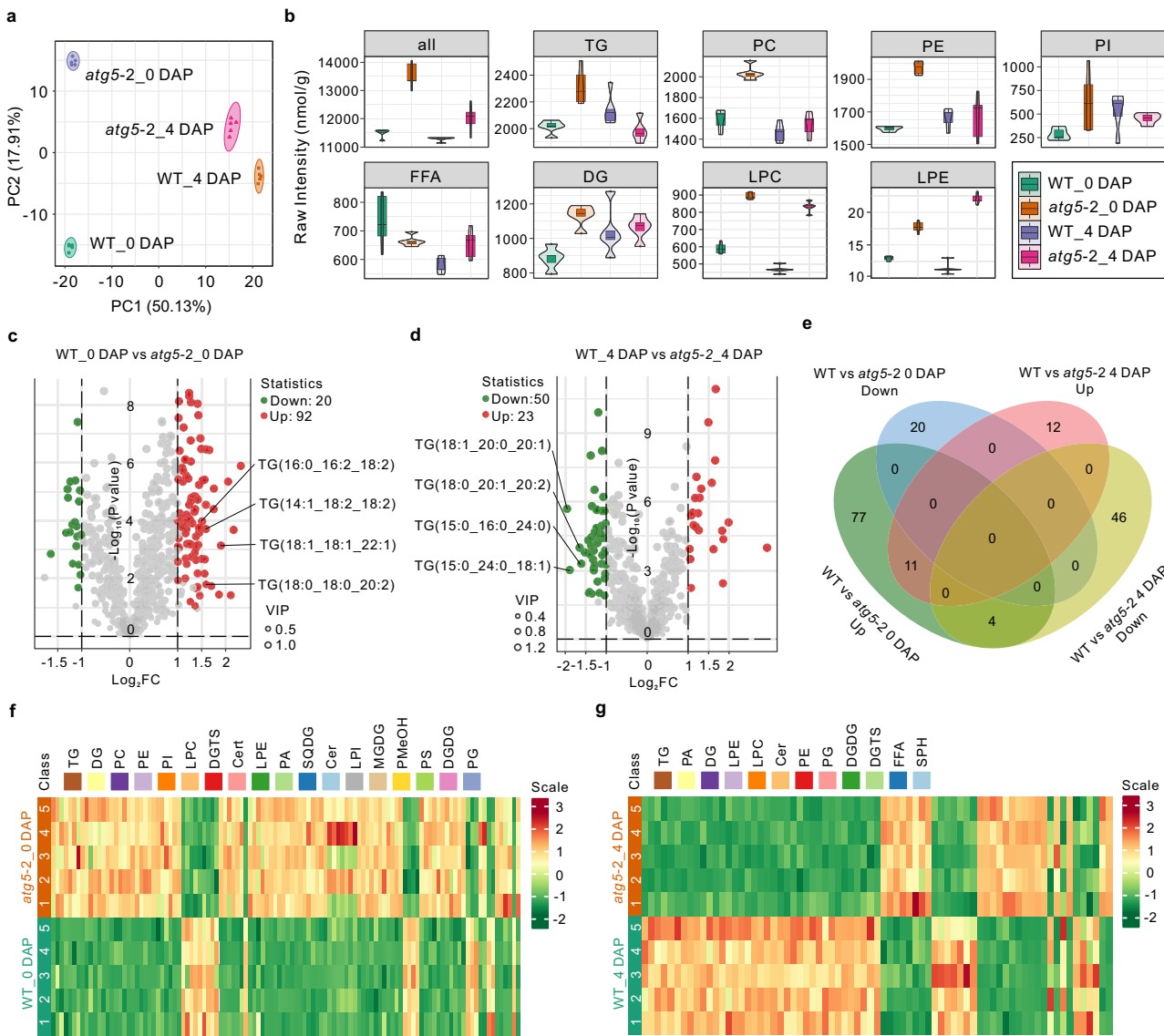

**Fig. 7 | Impact of autophagy deficiency on lipid metabolism during seed development. a** Principal component analysis of the lipidomic data sets from the ovules and 4-DAP seeds in WT and *atg5-2* mutant. *n* = 5 Biologically independent replicates. **b** Violin plots display the content of various lipid subclasses from ovules and 4-DAP seeds in WT and *atg5-2* mutants. The black horizontal line right in the middle represents the median, the middle box represents the quartile range, the thin black line extending from it represents the 95% confidence interval, and the outer shape represents the distribution density of the data. *n* = 5 Biologically independent samples. **c**, **d** Volcano plots of differential metabolites between WT and *atg5-2* ovules (**c**) or between the WT and *atg5-2* seeds at 4 DAP (**d**), with significantly up-regulated metabolites shown as red dots and significantly down-

regulated metabolites shown as green dots; the size of the dots represents VIP values. VIP variable importance in projection, FC fold change. **e** Venn plot comparing differential metabolites between the WT and *atg5-2* ovules, and the WT and *atg5-2* seeds at 4 DAP. **f, g** Heat maps displaying the relative lipid contents from the WT and *atg5-2* ovules (**f**) and seeds at 4 DAP (**g**). TG triacylglycerol, DG diacylglycerol, FFA free fatty acid, PC phosphatidylcholine, PE phosphatidylethanolamine, PI phosphatidylinositol, PA phosphatidic acid, PS phosphatidyl serine, PG phosphatidyl glycerol, LPC lyso-PC, LPE lyso-PE, LPI lyso-PI, Cer ceramide, Cert phytoceramide, MGDG monogalactosyl diacylglycerol, PMeOH phosphatidylmethanol, DGDG digalactosyl diacylglycerol, SPH sphingosine, DGTS diacylglyceryl-trimethylhomoserine, SQDG sulfoquinovosyl diacylglycerol.

(Fig. 7c) or between WT and *atg5-2* 4-DAP seeds (Fig. 7d), representing the consequences of autophagy on lipid metabolism during early seed development. We identified only 11 lipid species commonly upregulated by autophagy at different developmental stages (Fig. 7e), suggesting that autophagy plays a distinct developmental-stage-dependent role in lipid metabolism. The levels of TGs were significantly higher in *atg5-2* ovules than in the WT, whereas we observed decreased TG levels with a concomitant increase in free fatty acids in *atg5-2* seeds at 4 DAP relative to WT seeds (Fig. 7b, f, and g), suggesting a dual role for autophagy in lipid metabolism during early seed development.

As mentioned above, autophagy-mediated degradation of the integumentary tapetum is critical for seed development. We thus focused on the role of autophagy in integumentary tapetum lipid metabolism. Consistent with the lipidomic analysis, BODIPY 493/503 staining for neutral lipids in ovules and 4-DAP seeds demonstrated that autophagy contributes to TG metabolism in the integumentary tapetum. Indeed, we observed higher TG contents in both *atg5-2* and *atg7-1* ovules before fertilization, but lower TG levels in the seeds after fertilization (Fig. 8a–e), suggesting that autophagy has a distinct role in the lipid metabolism of the integumentary tapetum during early seed development.

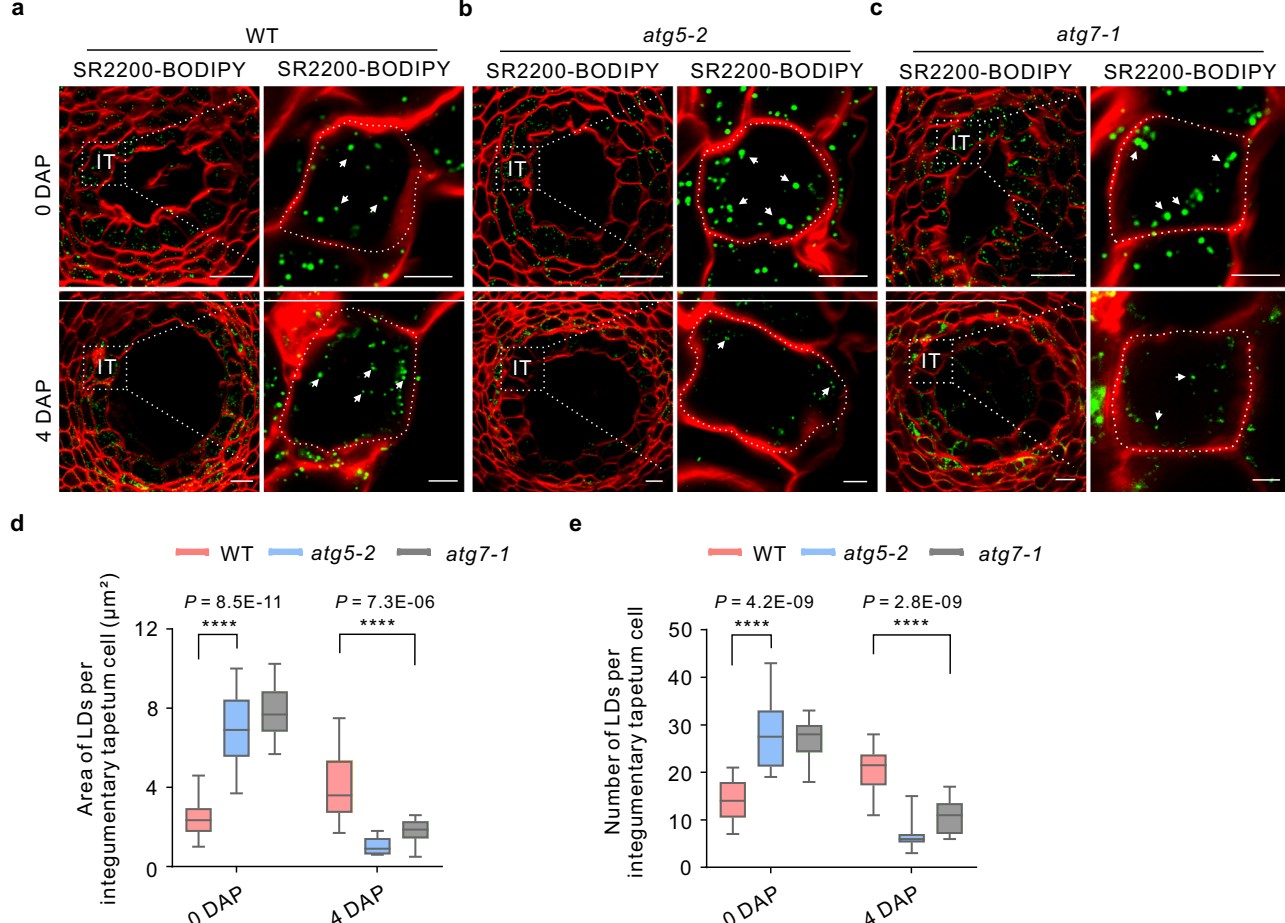

**Fig. 8 | Altered triacylglycerol (TG) levels in the integumentary tapetum of autophagy-deficient mutant. a–c** Lipid droplets in the integumentary tapetum from the WT (**a**), *atg5-2* (**b**), and *atg7-1* (**c**) mutants were stained with BODIPY 493/503. Arrows indicate lipid droplets. IT integumentary tapetum, DAP days after pollination, SR2200 SCRI Renaissance 2200. Scale bars: 20 μm (left); 5 μm (right).

**d**, **e** Quantitative analysis of the area (**d**) and number (**e**) of lipid droplets within the integumentary tapetal cells of WT, *atg5-2*, and *atg7-1* mutants. The data are displayed in the box and whisker plots. Center line, median values; top and bottom of boxes, 75th and 25th percentiles; top and bottom lines, maximum and minimum values (*n* = 20 independent samples), (two-tailed Student's *t* test, ****P < 0.0001).

## Discussion

Autophagy is an important pathway for the degradation of intracellular components and has been implicated in PCD[47–49]. PCD in plants can be further divided into environment-induced PCD (ePCD) and developmental PCD (dPCD)[50]. Whereas the relationship between autophagy and ePCD is relatively well understood, the role of autophagy in dPCD is still largely unknown. One recent study reported that this role is cell type-specific: autophagy is necessary for timely dPCD in proximal root cap cells, but dispensable for dPCD execution of distal lateral root cap cells, suggesting a cell death promotion role for autophagy in dPCD in proximal root cap cells[14]. It has also been suggested that autophagy is involved in dPCD of suspensor cells in somatic embryos[51], as well as in the tracheary elements of the xylem[52] and the anther tapetum[8], implying a common role for autophagy in promoting dPCD in different cell types.

Timely PCD of the anther tapetum, the counterpart of the integumentary tapetum, is essential for microsporogenesis and male gamete fertility. Tapetum PCD defects result in the formation of abnormal pollen grains, hence, male sterility[24,53]. In rice (*Oryza sativa*), for example, a mutation in *TAPETUM DEGENERATION RETARDATION* (*TDR*), encoding a basic helix−loop−helix transcription factor, delays tapetum PCD, leading to aborted pollen development and male sterility[24]. Autophagy also contributes to PCD in the rice anther tapetum. Indeed, mutation in *ATG7* leads to tapetum degradation defects, causing complete sporophytic male sterility, indicating that

autophagy-mediated anther tapetum PCD is crucial in male gametophyte development[8]. Here, we demonstrated that the integumentary tapetum also undergoes degeneration during seed development in tobacco, displaying typical characteristics of plant dPCD. Our present results are consistent with the involvement of autophagy in dPCD, demonstrating significant autophagic activities in the integumentary tapetum that promote dPCD during seed development. Blocking autophagy by mutation of autophagy-specific genes, such as *ATG5* or *ATG7*, resulted in delayed PCD in the integumentary tapetum, indicating that autophagy plays a critical role in promoting the initiation of dPCD in the integumentary tapetum.

Seed formation is a complex program that entails the growth of three genetically distinct entities: the seed coat, the embryo, and the endosperm[31]. The integumentary tapetum is a layer of cells specified from the innermost integuments that surround each developing embryo sac in flowering plants. Integumentary tapetal cells exhibit cytological characteristics similar to those of secretory cells and share similar features with the anther tapetum[19]. The anther tapetum acts as a safeguard for the developing pollen grains by providing them with essential nutrients, enzymes, and precursors for pollen development[23,54]. However, the role of the integumentary tapetum in seed development has long been overlooked. It has been speculated that this tissue serves multiple roles in seed development, such as providing the necessary nutrients and supporting embryo and endosperm development, based on its strategic location and cytological

characteristics. Consistent with this notion, a recent study revealed the significance of symplastic signals through plasmodesmata of integuments for the pre-fertilization development of the embryo sac[55]. In addition, another recent research demonstrated that micropylar integumentary-tapetum-generated gibberellin serves as a maternal signal regulating PCD initiation in the embryonic suspensor through the NtGID1-NtCRF1-NtCYS-NtCP14 signaling cascade[56]. Gibberellin transiently increases in abundance in the micropylar integumentary tapetum after fertilization and is transported to the basal suspensor cell, where it regulates the module responsible for suspensor PCD. Studies on the cysteine protease gene *NtTPE8* specifically expressed in integumentary tapetum cells and the serine proteinase inhibitor II gene *SaPIN2* from American black nightshade (*Solanum americanum*) also support a critical role for integumentary tapetum in early embryonic development[25,26]. Silencing of *NtTPE8* resulted in embryo abortion despite the exclusive expression of this cysteine protease gene in the integumentary tapetum, suggesting a non-cell-autonomous role for the integumentary tapetum in regulating embryo development[25]. In the present study, we discovered that timely PCD in the integumentary tapetum is crucial for correct embryo pattern formation, similar to the role of anther tapetum PCD in microsporogenesis. Specific expression of *ATG5* in the integumentary tapetum partially rescued the defects in embryo pattern formation, suggesting that cell-to-cell communication between the seed coat and the embryo is crucial for proper embryo development. Elucidating the connections between integumentary tapetum and embryo in further studies will facilitate our understanding of the mechanisms of early seed development.

Research in animal systems has revealed an intricate interplay between autophagy and lipids[41,44]. TGs stored in lipid droplets can be degraded through a form of selective autophagy called lipophagy. Suppressing autophagy significantly increases TG storage in lipid droplets[41]. In addition to lipid degradation, autophagy appears to also be implicated in TG biogenesis, although the exact mechanisms are still largely unknown[57]. Compared to the amount of research conducted on animals, the amount of research on the connection between autophagy and lipids in plants lags behind[58–60]. Similar to animals, recent evidence also suggests a dual role of autophagy in lipid metabolism, involving participation in triglyceride degradation and biogenesis, which is exhibited not only under different growth conditions but also at different developmental stages[61,62]. In Arabidopsis, blocking basal autophagy activities by mutating *ATG2* or *ATG5* led to a significant decrease in TG content in mature seeds and 4-day-old seedlings, whereas lipid droplets were degraded through starvation-induced lipophagy, suggesting different roles for autophagy in lipid metabolism between basal autophagy under normal development and stress-induced autophagy under stress conditions[61]. Similarly, in Chlamydomonas, a recent study suggests distinct roles of autophagy in lipid metabolism at different starvation stages, with its potential contribution to lipid droplet formation during the early stages and likely involvement in lipid droplet degradation during the later stages[62]. Our present study revealed that the role of autophagy in lipid metabolism during integumentary tapetum development is dependent on the developmental stage in tobacco. Autophagy-deficient mutants experienced a significant increase in the TG contents of their ovules before fertilization, whereas we observed a marked decrease in TG contents in *atg5* mutant seeds after fertilization. This finding uncovers a previously underappreciated developmental-stage-dependent role for autophagy in lipid metabolism. Additionally, the role of autophagy in lipid metabolism is also likely cell type and development dependent. This idea is supported by the lower TG contents observed in the mature anthers of the *Osatg7* mutant[8], whereas the abundance of TGs increases significantly in autophagy-deficient pollen grains[9]. Taken together, the role of autophagy in plant lipid metabolism is complex, as it plays a cell-type- and developmental-

stage-dependent role under normal growth conditions. However, the regulatory mechanisms regarding the distinct roles of autophagy in lipid biosynthesis or degradation in different cell types are worthy to be elucidated in further studies.

## Methods

### Plant materials
*N. tabacum* L. cv. Petite Havana SR1 plants were grown under 12 h light/12 h dark at 25 °C in a glasshouse.

### Vector construction and plant transformation
To generate *proATG5:H2B-GFP* and *proATG7:H2B-GFP* constructs, the promoter sequences of *ATG5* and *ATG7* were amplified from tobacco genomic DNA and inserted into the pART27 vector upstream of the *H2B-GFP* cassette. CRISPR/Cas9-based genome editing system was introduced to generate *atg5* and *atg7* mutants. The gRNAs for *ATG5* and *ATG7* were designed and cloned into the pYLCRISPR/Cas9P35S-N according to the previous protocol[63]. Briefly, suitable target sites were selected, and then PCR primers were designed according to the target sites. Next, the target sequences were introduced into corresponding sgRNA expression cassettes through PCR. Finally, the sgRNA expression cassettes were inserted into the pYLCRISPR/Cas9P35S-N vector. To generate *proATG5::ATG5-GFP* and *proTPE8:ATG5-GFP* constructs, the coding sequences of *ATG5* amplified from the cDNA prepared from the seeds and cloned into the pART27 vector upstream of GFP to generate *ATG5-GFP* fusion constructs. The promoters of *ATG5* and *TPE8* were inserted upstream of *ATG5-GFP* to generate *proATG5:ATG5-GFP* and *proTPE8:ATG5-GFP*, respectively. To generate *proTPE8:GFP-ATG8*, the *ATG8* coding sequence was amplified and cloned into the pART27 vector downstream of GFP, and then the promoter of *TPE8* was inserted upstream of *GFP-ATG8*. Primers for vector construction are listed in the Supplementary Table 1. *Agrobacterium tumefaciens*-mediated transformation of tobacco was performed to generate transgenic plants.

### Immunofluorescence
Ovules and seeds from WT and *atg* mutants were collected to prepare sections for subsequent immunofluorescence assay according to the previous protocol with minor modifications[64]. Briefly, the ovules and seeds were fixed in 4% paraformaldehyde for 12 h and then washed three times with PBS. After the fixation, the samples were dehydrated using an alcohol gradient from 10% to 100% and replaced with a gradient of xylene from 25% to 100%. The samples were then embedded under a series of paraffin xylene mixtures. Sections at 10 μm were prepared using a rotary microtome (Leica RM2245, Germany). After de-paraffinization, rehydration, and antigen retrieval, the sections were then used for immunofluorescence. The slides were incubated in 5% BSA solution at 4 °C for 12 h. The samples were then washed and incubated with the anti-ATG8A antibody (Abcam, ab77003; 1:100 dilution) dilution or anti-ATG7 antibody (produced by Abclonal, E3633; 1:100 dilution) for 1 h at room temperature. After washing with PBS three times, the samples were stained with an Alexa Fluor 488-conjugated goat anti-rabbit secondary antibody (1:200 dilution; Abcam, ab150077) for 2 h at room temperature. For double immunofluorescence, ovules from *proTPE8:GFP-ATG8* plants were collected, and paraffin sections were prepared as described above. The slides were then incubated with both the anti-ATG8A antibody (Abcam, ab77003; 1:100 dilution) and the anti-GFP antibody (DIA-AN, 2057; 1:100 dilution) for 1 h at room temperature. After washing with PBS three times, the samples were stained with an Alexa Fluor 594-conjugated goat anti-rabbit secondary antibody (Thermo Fisher Scientific, A-11012; 1:200 dilution) and an Alexa Fluor 488-conjugated goat anti-mouse secondary antibody (Thermo Fisher Scientific, A-11001; 1:200 dilution) for 2 h at room temperature. The samples were observed under a confocal laser scanning microscope (CLSM) (Leica

TCS SP8, Germany) with the excitation at 488 nm/emission at 500 ± 20 nm for Alexa Fluor 488 and excitation at 561 nm/emission at 600 ± 20 nm for Alexa Fluor 594.

## TUNEL assay

Sections of ovules and seeds were prepared following the protocol described above for immunofluorescence assay. DNA fragmentation was detected using the DeadEnd™ Fluorometric TUNEL System (Promega, G3250) according to the manufacturer's instructions. In brief, the paraffin sections were dewaxed with xylene, rehydrated with ascending grades of alcohols, fixed in 4% paraformaldehyde for 15 min, and washed with PBS. Subsequently, the samples were digested with 20 μg/mL Proteinase K for 10 min, fixed again in 4% paraformaldehyde for 5 min, washed with PBS, treated with Equilibration buffer for 10 min, and incubated in 100 μL rTdT incubation buffer at 37 °C for 60 min. Finally, the reaction was terminated in 2× SSC solution for 15 min, and the samples were washed by fresh PBS. The samples were then observed under a confocal microscope (Leica TCS SP8, Germany) under excitation with a 488 nm laser, and emission measured at 520 ± 20 nm.

## BODIPY staining

The ovules and seeds were collected and embedded in the low-melting-point agarose (Promega, V2111). The fresh sections of ovules and seeds were prepared with the fully automated Leica VT1200 S vibrating blade microtome (Leica, Germany) for BODIPY staining. Fresh sections were then stained with the BODIPY 493/503 (Thermo Fisher Scientific, D3922) and SCRI Renaissance 2200 (SR2200) for 10 min. After washing with PBS three times, the sections of ovules and seeds were observed under a CLSM (Leica TCS SP8, Germany) with the excitation at 405 nm/emission at 435 ± 20 nm for SR2200 and excitation at 488 nm/emission at 515 ± 15 nm for BODIPY.

## CLSM observation and image analysis

Samples observation and image acquisition were performed by a CLSM (Leica TCS SP8, Germany). To observe GFP fluorescence in the ovules and seeds of *proATG5:H2B-GFP*, *proATG7:H2B-GFP*, *proTPE8:GFP-ATG8*, and *proTPE8:ATG5-GFP* transgenic plants, the fresh sections were prepared using the fully automated Leica VT1200 S vibrating blade microtome (Leica, Germany) for CLSM observation. Samples were excited at 552 nm laser with emission measured at 620 ± 25 nm for PI observation, and excited at 488 nm laser with emission measured at 520 ± 20 nm for GFP observation. The images used for quantitative comparisons were acquired under identical parameter settings. Image processing was performed using the Leica Application Suite X (LAS X 3.5.5) and ImageJ (1.53k).

## TEM analysis

The procedure for TEM assay was performed according to our previous protocol with minor modifications[9]. Samples were embedded through high-pressure freezing and freeze substitution. For high-pressure freezing, isolated ovules and seeds were collected into the cryoprotectant (Promega, V2111), transferred to the carrier, and then fixed by a high-pressure freezing system (Leica EM HPM100). Freeze substitution was then carried out using a freeze substitution processor (Leica EM AFS2) with acetone containing 2% osmic acid for a 5-day workflow as follows: −90 °C for 72 h, 3.75 °C per hour raise to −60 °C and lasts 12 h, 3.75 °C per hour raise to −30 °C and lasts 12 h, 3.75 °C per hour raise to 0 °C and lasts 2 h. After extensive washing with acetone, the samples were then embedded under a series of Spurr's resin acetone gradient mixtures (30–100%). The embedding block was trimmed by a milling system (Leica EM TRIM2), and samples were sectioned into 70 nm ultra-thin slices with an ultramicrotome (Leica EM UC7). After staining with 2% uranyl acetate, the samples were observed with a TEM (JEM-1400, JEOL) equipped with the supporting camera system DS-L1 (Gatan).

## Lipidome analysis

Ovules and 4-DAP seeds from both WT and *atg5* plants were collected for lipid extraction and subsequent lipidome profiling. The extraction and lipidome profiling was carried out using a previously published protocol[64]. All samples were ground in liquid nitrogen and freeze-dried using SCIENTZ-100F. Then, the lipid was extracted by a solvent system of methanol and methyl tert-butyl ether (MTBE). In detail, 1 mL of lipid extract (MTBE:MeOH = 3:1) was added to 20 mg of freeze-dried sample and vortexed for 30 min. Subsequently, 300 μL of ultra-pure water was added and vortexed for 1 min. The supernatant was obtained by centrifugation and concentrated to dryness at 20 °C. Finally, 200 μL of a lipid solution (CAN:IPA = 1:1) was added to redissolve the dried extract. During the experiment, we used a mixture of test samples as the quality control sample. We analyzed a total of four quality control samples during the experiment.

The data acquisition instrument system mainly includes Ultra Performance Liquid Chromatography (UPLC, ExionLC™ AD, SCIEX) and Tandem Mass Spectrometry (MS/MS, QTRAP® 6500+, SCIEX). Liquid phase conditions are listed as follows: column, Thermo Accucore™C30; mobile phase A, acetonitrile/water, 60/40 (V/V), 0.1% formic acid, 10 mmol/L ammonium formate; mobile phase B, acetonitrile/isopropanol, 10/90 (V/V), 0.1% formic acid, 10 mmol/L ammonium formate; gradient elution program, A/B (V/V), 80/20 at 0 min, 70/30 at 2 min, 40/60 at 4 min, 15/85 at 9 min, 10/90 at 14 min, 5/95 at 15.5 min, 5/95 at 17.3 min, 80/20 at 17.5 min, and 80/20 at 20 min; flow velocity, 0.35 mL/min; column temperature, 45 °C; sample volume, 2 μL. The effluent was alternatively connected to an ESI-triple quadrupole-linear ion trap (QTRAP)-MS. LIT and triple quadrupole scans were obtained on a triple quadrupole-linear ion trap mass spectrometer (QTRAP® 6500+ LC–MS/MS System), equipped with an ESI Turbo Ion–Spray interface, operating in 500 °C source temperature, 5500 V positive ion spray voltage, −4500 V negative ion spray voltage, 45 psi ion source gas 1, 55 psi gas 2, 35 psi curtain gas and controlled by Analyst 1.6.3 software. Quantification was performed using the multiple reaction monitoring (MRM) mode of a triple quadrupole mass spectrometer, with the MS1 mass resolution 3000–5000. Differential metabolites are determined by both VIP ≥ 1 and absolute $Log_2FC$ (fold change) ≥ 1. VIP values were extracted from OPLS-DA results, which were generated using the R package MetaboAnalystR.

## Embryo isolation

Living embryo isolation was performed according to our previously established protocol[34,65]. Briefly, seeds at 6 DAP were treated with an enzyme solution comprising 1% Cellulase R-10 and 0.8% Macerozyme R-10 (Yakult Honsha Co. Ltd, Tokyo, Japan) dissolved in 11% mannitol for 30 min. Embryo sacs containing 8-cell embryos were first released by gentle grinding under an inverted microscope. Subsequently, 8-cell embryos were mechanically isolated from the embryo sacs after a second enzymatic treatment for 10 min with 0.25% Cellulase R-10 and 0.2% Macerozyme R-10 dissolved in 11% mannitol. Following extensive washing with 11% mannitol three times, the isolated 8-cell embryos were collected for subsequent mRNA extraction and library construction.

## mRNA extraction and RNA-seq library construction

mRNA extraction and cDNA preparation were performed according to our previous protocol[66]. mRNA extraction was carried out using the Dynabeads mRNA DIRECT Micro Kit (Life Technologies, USA), and cDNA preparation was performed using SMART-seq v4 Ultra Low Input RNA Kit (Takara Bio USA). cDNA was purified using AMPure XP beads (Beckman Coulter, A63880). Purified cDNA was then quantified using

the Agilent 2100 Bioanalyzer (Agilent Technologies, USA). RNA-seq libraries were prepared using the VAHTS Universal Plus DNA Library Prep Kit for MGI (Vazyme Biotech) and then sequenced on an MGI DNBSEQ-T7 with a 2 × 150 bp paired-end model.

## RNA-seq data analysis

The raw reads were filtered using fastp with the option "--detect_a-dapter_for_pe" to eliminate low-quality reads and adapters[67]. Subsequently, the clean reads were aligned to the *N. tabacum* reference genome using STAR (v.2.7.11a) with the parameter "--outSAMtype BAM SortedByCoordinate"[68]. The reads count and fragments per kilobase of transcript per million mapped reads (FPKM) were calculated using RSEM (v.1.3.3) with "--paired-end --no-bam-output"[69]. Differential gene expression analysis was conducted using the DESeq2 package[70] according to the following criteria: fold change ≥ 2; Benjamini−Hochberg adjusted *P* value < 0.05. K-means cluster analysis of DEGs was performed using the ClusterGVis package (https://github.com/junjunlab/ClusterGVis). GO enrichment analysis were performed on the PANTHER website (https://pantherdb.org/) with "Fisher's exact test" and "Calculate False Discovery Rata". The plots were generated using the R package ggplot2 (v.3.4.3).

## Immunoblot assay

To characterize ATG protein levels in WT and *atg* mutants, total proteins were extracted using the buffer composed of 50 mM Tris-HCl (pH 8.0), 150 mM NaCl, 1 mM EDTA, 1 mM phenylmethylsulfonyl fluoride, 10 mM iodoacetamide, and proteinase inhibitor cocktail (Roche, 4693132001). The extracts were then subjected to SDS-PAGE and subsequently transferred to nitrocellulose membrane (GVS North America, 1212590) for immunoblotting with rabbit anti-ATG5 antibody (produced by Abclonal, E3631; 1:2000 dilution) or rabbit anti-ATG7 antibody (produced by Abclonal, E3633; 1:2000 dilution)[9]. ATG8−PE assay was performed according to the previous protocol[9]. Briefly, the extracts were subjected to SDS-PAGE on 12% polyacrylamide gels in the presence of 6 M urea (Sigma-Aldrich, U6504). The separated proteins were then transferred to nitrocellulose membranes (GVS North America, 1212590), and probed with anti-ATG8A antibody (Abcam, ab77003; 1:1000 dilution). Antibody against β-actin (Abbkine, A01050; 1:2000 dilution) was used as the loading control in western blot analysis. HRP Goat Anti-Mouse IgG (Abclonal, AS003; 1:4000 dilution) or HRP Goat Anti-Rabbit IgG (Abclonal, AS014; 1:4000 dilution) were used as secondary antibodies in the immunoblot assay.

## VPE activity assay

VPE proteolytic activity assay was performed according to the previous protocols with minor modifications[38]. Briefly, total proteins were extracted from seeds at various stages using an extraction buffer composed of 50 mM MES (pH 6.0), 2 mM EDTA, 10% glycerol, 0.1% CHAPS, 0.01% Brij-35, 2% polyvinylpolypyrrolidone, 10 mM L-cysteine, and 10 mM sodium metabisulphite. Proteolytic activities were subsequently measured in a reaction buffer consisting of 50 mM sodium acetate (pH 5.5), 10 μg of total protein, 50 μM substrate Z-AAN-AMC (Bachem, 4033201), 10 mM L-cysteine, 1 mM EDTA, and 0.01% Brij-35. The fluorescence levels of the released AMC were monitored using a SPECTRAmax iD5 microplate reader (Molecular Devices, USA) with an excitation wavelength of 360 nm and an emission wavelength of 455 nm.

## Plot preparation and statistical analysis

Bar−dot and box−whisker plots were prepared using GraphPad Prism 9. Two-sided Student's *t* test and Tukey−Kramer multiple comparison test were performed for statistical analysis.

## Reporting summary

Further information on research design is available in the Nature Portfolio Reporting Summary linked to this article.

## Data availability

Lipidomics data are provided with this paper. RNA-seq data have been uploaded to the NCBI Gene Expression Omnibus (GEO) under accession GSE248624. All seeds and other materials related to the findings of this study are available from the corresponding authors upon reasonable request. Source data are provided with this paper.

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

## Acknowledgements

This work was supported by National Natural Science Foundation of China (31991201, 31970340), the Major Project of Hubei Hongshan Laboratory (2022hszd017), and the Science and Technology Department of Hubei Province (2022CFA071). We thank Prof. Y.G. Liu (South China Agricultural University) for the kind offer of the CRISPR/Cas9 vectors.

## Author contributions

P.Z. and M.-x.S. designed the experiments. L.-l.Z., R.C., Z.B., J.L., and Y.Z. performed the experiments. L.-l.Z., Y.Z., P.Z., and M.-x.S. contributed to the data analysis and wrote the manuscript.

## Competing interests

The authors declare no competing interests.
