## [Peer Review File · Nature Communications]

Autophagy-Mediated Degradation of Integumentary Tapetum Is Critical for Embryo Pattern FormationReviewer #1 (Remarks to the Author):

General Comments: The manuscript is well organized and written. Zhao et al. report that autophagy is crucial for timely PCD in the integumentary tapetum, influencing embryo pattern formation and seed viability. They highlight the critical role of autophagy in regulating timely integumentary tapetum PCD and reveals its significance in seed lipid metabolism and viability. Generally, the quality of the data is technically sound, obtained with appropriate techniques, analysed and interpreted carefully. This manuscript presents an interesting story, and reveals the molecular mechanism of autophagy-mediated degradation of integumentary tapetum and its effects on embryo pattern formation. They provide relatively strong evidences and appropriate controls. These results will be important to the field and advance our understanding on embryo pattern formation and early seed development. However, there are still some minor issues which should be addressed before it can be considered to be published in Nature Communications.

Detailed Comments:

1. In lines 43-44, the incited reference of autophagy involving in pollen and anther development lost some important ones, such as, Zhu et al., Normal Structure and Function of Endothecium Chloroplasts Maintained by ZmMs33-mediated Membrane Lipid Biosynthesis in Tapetal Cells Are Critical for Anther Development in Maize. *Mol Plant*, 2020, 13: 1624–1643
2. In lines 97-101, why autophagosomes through ATG8 immunofluorescence were used as control? It should be explained and added a reference here.
3. In lines 180-182, a reference of TUNEL assay should be incited here.
4. In lines 277-278, why autophagy affects the developmental-stage-dependent lipid metabolism during early seed development? It should be provided more evidences and discussed in more detailed in the DISCUSSION part.

Reviewer #2 (Remarks to the Author):

The manuscript by Zhao et al describes how the activity of the ATG genes (particularly ATG5) is key for proper endothelium (integumentary tapetum, IT) development and PCD, which in turn has also a role in the correct embryo and seed development. Data seems to be convincing, although a direct link between ATG function and embryo development was not provided.

-It is not clear what is the relative contribution of each ATG gene to endothelium degradation. They use ATG5 and its mutants to clearly show that it is involved. ATG7 could also be involved although not comprehensive analysis with this gene/mutants is shown. Is there any other ATG genes involved? Why did they start working with ATG5 and ATG7? They have previous data that indicate the most ATGs are expressed in developing seeds (lines 83-84).

-A major issue is the lack of any hint toward the molecular mechanism that links PCD of the endothelium with embryo development. Perhaps, by the use some of the previous data on this matter. For example, to check whether atg mutants alter local GA synthesis in micropylar integumentary tapetum (reference #41). This is quite relevant as the paper in reference #41 is by the same authors, so they have all molecular tools to test this.

Based on previous work it seems that cell-to-cell communication between integuments and the embryo is critical (see the very recent paper in *Plant Journal* by Qin et al. on plasmodesmata communication with the embryo sac). References to this sort of communication is worth to be mentioned in Discussion.

-A weak point is the lipidome analysis, as in a whole it has little relationship with the rest of the paper.

-Another issue is the use of ATG8 as marker. It is not well justified. Does it have a role in endothelium autophagy?

-A major issue is the quality of the images, as this paper relies mostly on the ovule/seed images

shown in most figures. To me, it is not clear that medium plane images are taken in all cases, as the images show quite differences that suggests that different planes are used.

In addition, and very important, the scale in the figures seem to be different. For example, in Figure 3, the scale for images at 4DPA between WT and *atg5* mutants is very different, meaning that the mutant seed is much bigger than WT, which this is not the case for 2 or 6 DPA. Moreover, just by looking at the images, it is not clear that the endothelium is wider in mutant than in the WT.

-In the introduction, there is a lack of information regarding the endothelium; more information on the importance of the endothelium should be provided. How does its degradation proceed? Was it analyzed previously? For example, it would be necessary to indicate when the IT is degraded after fertilization. This would also be important to interpret data in Figure 1.

-Line 100. Explain where TPE is expressed and where its promoter drives expression.

-Data related to the percentage of aborted seeds is not very precise. I wonder if they have data on total number of seeds per fruit instead.

-When does the IT degradation occur? Why did they use 0 and 2 DPA in Figure 1A and B? are these key developmental stages?

-It is also important mentioning that both ATG5 and ATG7 are expressed at higher levels in another cell layers on ovules, rather than in the endothelium (GFP signal in Figure 1A and B. Justify this.

-It is not clear what we are looking at Figure 1C. The zoom out area is not clear to me for the PI and anti-ATG8 images. I recommend that they use a better way to highlight the nucleus and the autophagosome in all panels, not only in the merge image. Also, draw a dotted line delimiting the endothelium as in figure 1 A-B or D.

-Are autophagosomes only visible in the endothelium (Figure 1D)? If so, mention this in the text.

-Remove "suggesting that it may play essential roles in early seed development" in lines 107-108.

-Did they target CRISPR to any specific domain of the ATG protein? Why did they target these exons? Please, provide some data on the characteristics of these mutants. Did they produce null alleles in all cases? They also must indicate the specific alleles used in each figure, not only a generic *atg5* or *atg7*.

-They should provide data on autophagosomes in *atg7* mutants, not only in *atg5*, as they use both to detect defects in seed development.

-Line 145 "exhibited delayed degradation". Does it mean that the endothelium in *atg5* mutant is finally degraded? They must show this data or rewrite the text.

-Lines 165-171. Data on Figure 3E is poorly described. It is not clear whether all embryos in *atg* mutants are defective, and are classified in three groups, or some embryos have normal WT morphology, therefore mutant seeds can be classified in 4 groups, the WT (normal embryos) plus 3 categories of defective embryos. This part of the text should be rewritten to properly describe the embryo phenotype of the mutant.

-Line 179-182. Indicate that TUNEL is used to detect fragmented DNA. Change the text as the TUNEL or TEM is not done only in integumentary tapetum cells, but rather to the whole seed ("we subjected the integumentary tapetum to...").

-Line 183-184. How is the TUNEL signal calculated. Please clarify what "52.2%; 47 out of 90" means.

-What are the two images in Figure 5A? It seems that the lower panel is the GFP signal and the

upper is the merged GFP plus the mPS-PI staining. Clearly indicate that in Figure and Figure legend.

-What does "Zygote" mean in Figure 5E? It seems that they refer to WT embryo/seed? Clarify and unify nomenclature with Figure 3E (where they refer to WT embryo).

-Line 226. To me, data resented does not "indicate" the existence of cell-to-cell communication. Indeed, it seems that ATG5 is important for endothelium development which has an impact in embryo and seed development, but it does not indicate a cell-to-cell communication. The last sentence in this paragraph must be rewritten to reflect the experimental data presented. They could indicate that their data may suggest a possible communication mechanism between the integumentary tapetum and the embryo.

Other Minor issues:

-Split and rewrite sentence in lines 141-144 "We observed clear developmental defects in the seed coat and the embryo in atg5 mutants. In WT seeds, the integumentary tapetum expanded from 0 to 4 DAP, and then gradually degraded. In contrast, the integumentary tapetum of atg5 mutant seeds ..."

-Line 167, introduce the Figure 3E at the end of this sentence, after the word "lineages", not several sentences later.

-It is quite difficult to differentiate the colors used for WT and atg5 in Figure 7C and D. Color blind readers will appreciate the use of other colors.

-The Figure legends should be unified in style and the way data is presented. For example, in most figures, scale bars are indicated in the corresponding panels but in Figure 7 it is indicated at the end of the legend, not after panels A and B.

-Reference 41 is introduced late. Move it to line 345.

-Line 328. Down-emphasize "orchestrates" in the title of this section. It seems clear that integumentary tapetum is important for embryo development, but I am not sure we can say that it "orchestrates" embryo development, unless more data is provided.

Reviewer #3 (Remarks to the Author):

Zhao et al., investigated the functions of autophagy in seed development, focusing on its centrality to the timely execution of programmed cell death (PCD) within the integumentary tapetum, analogous to the anther tapetum. Upon autophagy inhibition, a delay in the PCD of the integumentary tapetum was observed. Furthermore, it resulted in aberrations in embryo patterning. The targeted re-establishment of autophagic functions indicates the non-autonomous role of the integumentary tapetum in embryo patterning. Additionally, lipidomic analysis reveals distinct stage-dependent contribution of autophagy to seed lipid metabolism. The study finally reveals that autophagy affects triacylglycerol accumulation in integumentary tapetum, highlighting its criticality for seed viability. This study offers significant insights into the implications of

autophagy in the integumentary tapetum. However, I harbor several pivotal inquiries and concerns that need to be addressed. Additionally, I proffer a few recommendations that could potentially enhance the robustness of their findings. My detailed comments are delineated below:

Major concerns:

1. In Figure 1, the punctate dots depicted in Figure 1c are not sufficiently convincing as autophagosomes which are shown by the autophagosome reporter GFP-ATG8 shown in Figure 1d. A colocalization study using immunofluorescence labeling with the anti-ATG5 antibody in the integumentary tapetum expressing GFP-ATG8 would better support this conclusion. Additionally, Figure 1 lacks details on the subcellular localization and distribution of ATG7 in the integumentary tapetum.
2. In addition, the decreased autophagy level in the CRISPR/Cas9-generated *atg5* and *atg7* mutants needs verification in comparison to the WT. Simply calculating the number of fluorescent dots in the integumentary tapetum is not sufficiently convincing. To more effectively demonstrate the impairment of autophagy, a protein-level comparison between the mutants and the WT is also necessary.
3. A pressing question arising from this study is: How does autophagy influence cell patterning or distribution during embryo formation? What molecular links exist between autophagy and embryo pattern formation, and what are the underlying mechanisms? While the authors emphasize this in the title and abstract, supporting data is sparse with only Figure 3e directly addressing this. As such, the conclusions would be more convincing by benefiting from more supporting experimental data.
4. In Figure 4, the authors observed delayed PCD in the integumentary tapetum of the *atg5* mutant. Yet, the intricate relationship between integumentary tapetum PCD and autophagy remains ambiguous and warrants further evidence to solidify the conclusion. It's crucial to discern whether the delayed PCD in the integumentary tapetum directly or indirectly stems from impaired autophagy. Additionally, the mechanism by which autophagy might instigate this delayed PCD needs elucidation.
5. It is confusing that why the authors use the ovules and 4-DAP seeds from WT and *atg5* mutant for the lipid metabolism analysis especially since most prior observations and data relate to the integumentary tapetum. Considering seed development as a consequence of developmental defects in the integumentary tapetum, it seems more logical and convincing to analyze lipid metabolism directly within the integumentary tapetum. Doing so would provide unequivocal evidence of autophagy's role in regulating lipid metabolism in integumentary tapetum cells.
6. The mere observation of reduced triacylglycerol accumulation in the integumentary tapetum of *atg5* mutants seems cursory. It's essential to furnish evidence detailing how autophagy might directly influence this accumulation. How can we exclude the possibility that triacylglycerol accumulation isn't directly orchestrated by autophagy but rather manifests as a secondary effect in autophagy-deficient mutants?
7. The Introduction lacks comprehensive content. A concise overview of the biological roles of ATG5 and ATG7 in autophagy would be beneficial.

Minor concerns:

1. The letter "t" of Student's t test should be in italic.
2. Fig. 3a, the scale bars of 6 DAP are too small to be easily discerned.
3. Figure 4c lacks indicated scale bar lengths. In Figure 4d, scale bar lengths are only provided in the enlarged images, not in the original ones.
4. The red signal in Figure 5 is neither labeled nor described.

5. There's inconsistency in the reference style. For instance, names of mutants and species aren't italicized in the references (Lines 639, 656). They should be uniformed in reference style.

Response to Reviewer #1

Q1. General Comments: The manuscript is well organized and written. Zhao et al. report that autophagy is crucial for timely PCD in the integumentary tapetum, influencing embryo pattern formation and seed viability. They highlights the critical role of autophagy in regulating timely integumentary tapetum PCD and reveals its significance in seed lipid metabolism and viability.

Generally, the quality of the data is technically sound, obtained with appropriate techniques, analyzed and interpreted carefully. This manuscript presents an interesting story, and reveals the molecular mechanism of autophagy-mediated degradation of integumentary tapetum and its effects on embryo pattern formation. They provide relatively strong evidences and appropriate controls. These results will be important to the field and advance our understanding on embryo pattern formation and early seed development. However, there are still some minor issues which should be addressed before it can be considered to be published in Nature Communications.

Reply: Thank you for appreciation of our manuscript. Your comments have greatly helped us to improve the manuscript.

Q2. In lines 43-44, the incited reference of autophagy involving in pollen and anther development lost some important ones, such as, Zhu et al., Normal Structure and Function of Endothecium Chloroplasts Maintained by ZmMs33-mediated Membrane Lipid Biosynthesis in Tapetal Cells Are Critical for Anther Development in Maize. Mol Plant, 2020, 13: 1624–1643

Reply: The relevant references that you pointed out have been added to the revised manuscript.

Q3. In lines 97-101, why autophagosomes through ATG8 immunofluorescence were used as control? It should be explained and added a reference here.

Reply: In the manuscript, we employed both ATG8 immunofluorescence and the *proTPE8:GFP-ATG8* marker line as two independent approaches to visualize autophagosomes. This allowed for mutual verification of autophagic activity in the integumentary tapetum. Furthermore, we have included relevant references according to your suggestion.

Q4. In lines 180-182, a reference of TUNEL assay should be incited here.

Reply: A reference of TUNEL assay has been cited in the revised manuscript.

Q5. In lines 277-278, why autophagy affects the developmental-stage-dependent lipid metabolism during early seed development? It should be provided more evidences and discussed in more detailed in the DISCUSSION part.

Reply: Firstly, to confirm that autophagy regulates lipid metabolism in a developmental-stage-dependent manner, we also analyzed lipid metabolism in *atg7-1* mutants, focusing on triglycerides (TG) using BODIPY staining. Consistent with the results observed in *atg5-2* mutants, we observed higher TG contents before fertilization, but lower TG levels after fertilization in *atg7-1* mutants. New data have been added in the revised Figure 8. This

finding further confirms the developmental-stage-dependent role of autophagy in lipid metabolism during early seed development. In addition, we have included further discussions regarding the developmental-stage-dependent role of autophagy in lipid metabolism in the discussion section according to your suggestion.

Response to Reviewer #2

Q1. The manuscript by Zhao et al describes how the activity of the *ATG* genes (particularly *ATG5*) is key for proper endothelium (integumentary tapetum, IT) development and PCD, which in turn has also a role in the correct embryo and seed development. Data seems to be convincing, although a direct link between *ATG* function and embryo development was not provided.

Reply: Thank you for your comments, which have greatly helped us to improve the manuscript.

Q2. It is not clear what is the relative contribution of each *ATG* gene to endothelium degradation. They use *ATG5* and its mutants to clearly show that it is involved. *ATG7* could also be involved although not comprehensive analysis with this gene/mutants is shown. Is there any other *ATG* genes involved? Why did they start working with *ATG5* and *ATG7*? They have previous data that indicate the most *ATGs* are expressed in developing seeds (lines 83-84).

Reply: The pathways for autophagy are tightly regulated by a set of autophagy-related (*ATG*) genes that operate at different stages of autophagy. In plants, over thirty *ATG* genes have been identified, which can be categorized into five functional groups. These groups include the *ATG9* recycling system, the phosphoinositide-3-kinase complex, and two ubiquitin-like conjugation systems. Disrupting any of these functional systems through mutation of the *ATG* genes involved in these systems will result in defects in autophagy. Our previous results indicate that almost all *ATG* genes, which function at different steps of autophagy, are expressed in developing seeds. This observation is consistent with their respective roles in autophagy pathways and they are essential for the autophagic activity in early seeds. In the current study, we focused on two single-copy autophagy-specific *ATG* genes, *ATG5* and *ATG7*, to investigate the role of autophagy in integumentary tapetum degradation. Mutation in either *ATG5* or *ATG7* will lead to a deficiency in autophagy. In the revised manuscript, the role of *ATG7* in integumentary tapetum development has also been carefully analyzed, and new data have been included in (Figs. 2, 4, 8 and Supplementary Figs. 2, 4, 6). Although *ATG5* and *ATG7* act at different steps of autophagosome formation, they exhibit similar roles in regulating integumentary tapetum degradation. The results from *ATG5* and *ATG7* mutually confirm the role of autophagy in integumentary tapetum development. To clarify why we chose to work with *ATG5* and *ATG7*, we have incorporated additional information about autophagy as well as about the roles of *ATG5* and *ATG7* in autophagosome formation into the manuscript.

In fact, during the course of our experiments, we have also investigated the role of another *ATG* gene, *ATG10*, in seed development during the course of the experiment. Two independent *atg10* mutant lines were created using the CRISPR/Cas9 system. Consistent with observations in *atg5* and *atg7* mutants, approximately 30% of the seeds in the *atg10* mutant lines were aborted. The data about *ATG10* are listed in the Figure 1 below.

Fig.1 Mutation in *ATG10* results in seed abortion.

a *atg10* mutants were generated using the CRISPR/Cas9 system. Schematic diagrams displaying the genomic sequences of *ATG10*, with exons labeled by blue boxes, introns depicted by grey lines, and untranslated regions indicated by pink boxes. The inserted or deleted nucleotides are shown below the diagram. **b** Schematic diagrams showing the domains present in *ATG10* protein, along with the mutant versions generated through CRISPR/Cas9-edited mutagenesis. Missense sequences resulting from frameshift depicted as green boxes. **c** The frequencies of aborted seeds in *atg10* mutants. Data are from two independent experiments, with 8 ovaries for each experiment (n = 16). **** $P < 0.0001$ (Two-tailed Student's *t* test).

Q3. A major issue is the lack of any hint toward the molecular mechanism that links PCD of the endothelium with embryo development. Perhaps, by the use some of the previous data on this matter. For example, to check whether *atg* mutants alter local GA synthesis in micropylar integumentary tapetum (reference #41). This is quite relevant as the paper in reference #41 is by the same authors, so they have all molecular tools to test this.

Based on previous work it seems that cell-to-cell communication between integuments and the embryo is critical (see the very recent paper in Plant Journal by Qin et al. on plasmodesmata communication with the embryo sac). References to this sort of communication is worth to be mentioned in Discussion.

Reply: In response to your comments, we first attempted to test if gibberellic acid (GA) is the potential link between PCD in the endothelium with embryo development. Given that the generation of double homozygous plants by crossing the GA sensor marker line with *atg* mutants spans across at least three generations, and estimated to require an extensive period of about 2 years, we applied an alternative approach. We assessed the expression levels of GA biosynthesis and deactivation genes in WT and *atg5* mutants to test is there any difference in expression level of these genes. Our findings indicate that the expression of GA biosynthesis-related genes does not differ notably in the *atg5* mutants when compared to WT. This suggests that GA may not be the signaling molecule connecting PCD in the endothelium with the patterning of the embryo.

Secondly, to investigate the potential molecular links and mechanisms underlying the relationship between PCD in the integumentary tapetum and embryo pattern formation, we isolated embryos from the WT plants, *atg5* mutants, and *atg5* mutants expressing

proTPE8:ATG5-GFP (This approach aims to specifically restore *ATG5* expression in the integumentary tapetum while avoiding expression of *ATG5* in the *atg5* embryo.) for RNA sequencing. This transcriptomic data provides us with a good opportunity to investigate molecular connections between PCD of the endothelium and embryo development. Comparisons of differentially expressed genes between *WT/atg5-2* and *atg5-2 proTPE8:ATG5-GFP/atg5-2* demonstrated that the specific expression of *ATG5* in the integumentary tapetum leads to a significant decrease in the number of differentially expressed genes in the *atg5-2* embryos. This suggests that *ATG5* expression in the integumentary tapetum steers the embryonic gene expression profile in *atg5* mutants toward WT pattern. We then focused on the analysis of the set of genes in embryos that are regulated by autophagy in the integumentary tapetum. The results unveiled several well-known pathways, including the pattern specification process- and auxin signaling-related pathways that are associated with embryo pattern formation, regulated by autophagy in the integumentary tapetum. The results revealed that specific expression of *ATG5* in the integumentary tapetum could effectively rescue the defective gene expression profile of *atg5* embryos, further confirming the role of autophagy in embryo development. These insightful findings have been integrated into the new figure 6. In addition, we also included a discussion about cell-to-cell communication between integuments and the embryo in the revised manuscript, along with the relevant reference as you suggested.

Q4. A weak point is the lipidome analysis, as in a whole it has little relationship with the rest of the paper.

Reply: Recent researches demonstrate that autophagy and lipids have a close relationship (Singh et al., 2009; Fan et al., 2019; Barros et al., 2020). Lipids not only serve as regulatory factors for autophagy but also act as substrates of autophagy in diverse model systems. In animals, various lipid species such as phospholipids, glycerolipids, and sphingolipids are involved in different steps of autophagy, including autophagy initiation, autophagosome biogenesis, and maturation (Dall'Armi et al., 2013). In addition, autophagy also plays a role in regulating lipid metabolism, as storage lipids such as triglycerides (TG) can be degraded by autophagy (Singh et al., 2009). However, the relationship between lipids and autophagy in early seed development remains largely unknown. This study tried to investigate the relationship between autophagy and lipid metabolism during early seed development. To better integrate the lipidomic analysis section into the article, we have included several transition sentences in the revised manuscript.

Q5. Another issue is the use of *ATG8* as marker. It is not well justified. Does it have a role in endothelium autophagy?

Reply: *ATG8* is a core component of the autophagy machinery in both animals and plants. *ATG8* becomes conjugated to the lipid phosphatidylethanolamine (PE) through the ubiquitin-like conjugation system, enabling its association with the autophagosome membrane. These characteristics make *ATG8* an ideal marker for tracking the autophagy activity and it has been widely utilized as a marker for autophagosomes in both animals and plants (Chang et al., 2013; Gomez et al., 2022; Choi et al., 2023; Huang L. et al., 2024).

In the previous version of the manuscript, we assessed autophagic activities by

visualizing GFP-ATG8 and ATG8 immunofluorescence. In the revised manuscript, we evaluated the level of ATG8 conjugated with phosphatidylethanolamine (PE), known to be crucial for autophagosome formation, to confirm whether ATG8 could be used as a marker for monitoring autophagic activities. Consistent with the results of GFP-ATG8 visualization and ATG8 immunofluorescence, the level of lipidated forms of ATG8 (ATG8-PE) notably decreased in the *atg* mutants compared with that in the WT plants (Fig. 2h), further confirming that ATG8 is a reliable marker for tracking autophagy.

As discussed above, while ATG8 plays a conserved role in the autophagy process across various plant cell types and can be efficiently used as a marker to label autophagosomes, it is worthwhile to further confirm the role of *ATG8* genes in endothelium autophagy in the further studies. However, this process may require a considerable amount of time due to the existence of multiple ATG8 copy genes in plant genomes.

Q6. A major issue is the quality of the images, as this paper relies mostly on the ovule/seed images shown in most figures. To me, it is not clear that medium plane images are taken in all cases, as the images show quite differences that suggests that different planes are used.

In addition, and very important, the scale in the figures seem to be different. For example, in Figure 3, the scale for images at 4DPA between WT and *atg5* mutants is very different, meaning that the mutant seed is much bigger than WT, which this is not the case for 2 or 6 DPA. Moreover, just by looking at the images, it is not clear that the endothelium is wider in mutant than in the WT.

Reply: To compare the width of the endothelium between WT and *atg5* mutants, we employed two independent approaches: paraffin sections combined with toluidine blue staining and modified pseudo-Schiff propidium iodide whole mount staining. In both methods, we selected the median plane of a seed based on the maximum size of the embryo sac. For the modified pseudo-Schiff propidium iodide (PI) whole mount staining, successive sections of a seed after PI staining were checked from the up to the bottom using a confocal laser scanning microscope, and the section with the maximum size of the embryo sac was chosen to determine endothelium width. To provide a better description, a representative movie showing different sections of a seed has been submitted as Supplementary information (Supplementary Movies 1 and 2). In the case of paraffin sections combined with toluidine blue staining, successive sections of a seed were prepared and examined under a microscope, and the section with the maximum size of the embryo sac was also selected to determine endothelium width.

After careful checking the original figures, we noticed a careless mistake in the scale bar of the WT 4 DAP seeds when preparing the plane. Thank you for your careful reading, which helped us to identify and correct the error. Figure 4a has been revised accordingly. To further validate the results, we have carefully analyzed the endothelium of two additional independent *atg5* mutant lines and three independent *atg7* mutant lines. New data have been added in the supplementary figure 5 and 6, further confirming that the endothelium in autophagy-deficient mutants is wider than in the WT, demonstrating significant differences at 4 DAP and an expanded difference at 6 DAP.

In addition, we conducted a thorough review of the entire manuscript to assess the image

quality and improved figure preparation based on your suggestions.

Q7. In the introduction, there is a lack of information regarding the endothelium; more information on the importance of the endothelium should be provided. How does its degradation proceed? Was it analyzed previously? For example, it would be necessary to indicate when the IT is degraded after fertilization. This would also be important to interpret data in Figure 1.

Reply: Thank you for your suggestion. The revised manuscript now includes additional information about the endothelium during seed development, particularly focusing on its potential role and its degradation process during seed development.

Q8. Line 100. Explain where TPE is expressed and where its promoter drives expression.

Reply: *TPE8* is specifically expressed in the endothelium, and we used its promoter to drive GFP-ATG8 expression in the endothelium. The description of *TPE8* expression and relevant reference have been added to the revised manuscript.

Q9. Data related to the percentage of aborted seeds is not very precise. I wonder if they have data on total number of seeds per fruit instead.

Reply: The percentage of aborted seeds in both WT and *atg* plants was calculated based on statistical data from three independent replicates. For each replicate, seven ovaries were selected to determine the percentage of aborted seeds. Following your suggestion, we quantified the total number of seeds and aborted seeds for each ovary. The percentage of aborted seeds was then calculated based. All relevant data have been included in the revised manuscript (Figures. 2j and 5g).

Q10. When does the IT degradation occur? Why did they use 0 and 2 DPA in Figure 1A and B? are these key developmental stages?

Reply: In the present study, we observed that TUNEL signals were detected in the integumentary tapetal cells of 4-DAP seeds, but not in the 2-DAP seeds. This suggests that integumentary tapetum degradation occurs at 4 DAP. In addition, the developmental stages of the integumentary tapetum have also not been well characterized. In order to gain a better understanding of the developmental stages of the integumentary tapetum and its potential relationship with embryo development, we chose to investigate the integumentary tapetum based on key developmental stages of embryo development as described in our previous reports (Zhao et al., 2011; Zhao et al., 2013). For instance, 0 DAP represents unfertilized ovules, 2 DAP represents ovules after fertilization, 4 DAP represents seeds containing elongated zygote, 6 DAP represents seeds containing 8-cell embryos, and 8 DAP represents seeds containing globular embryos. These developmental stages are critical during seed development.

Q11. It is also important mentioning that both *ATG5* and *ATG7* are expressed at higher levels in another cell layers on ovules, rather than in the endothelium (GFP signal in Figure 1A and B. Justify this.

Reply: Based on previous expression analysis of *ATG* genes, the majority of *ATGs* are

widely expressed in various tissues, playing an important role in maintaining basal autophagic activities throughout plant development. The information about *ATG5* and *ATG7* expression in other layers of the ovules has been included in the revised manuscript according to your suggestion. This finding suggests basal autophagy is also active in the other layer of ovules.

Q12. It is not clear what we are looking at Figure 1C. The zoom out area is not clear to me for the PI and anti-ATG8 images. I recommend that the use a better way to highlight the nucleus and the autophagosome in all panels, not only in the merge image. Also, draw a dotted line delimiting the endothelium as in figure 1 A-B or D.

Reply: Following your suggestion, Figure 1C has been improved to clearly display the nucleus and the autophagosome within the endothelium. Moreover, in the revised version of Figure 1C, a dotted line has been added to delineate the endothelium.

Q13. Are autophagosomes only visible in the endothelium (Figure 1D)? If so, mention this in the text.

Reply: To better visualize autophagosomes in the endothelium, we employed an endothelium-specific promoter to drive GFP-ATG8 expression. As depicted in Figure 1D, autophagosomes were exclusively observed within the endothelium. The revised manuscript includes additional information.

Q14. Remove "suggesting that it may play essential roles in early seed development" in lines 107-108.

Reply: Based on your suggestion, the phrase "suggesting that it may play essential roles in early seed development" has been removed.

Q15. Did they target CRISPR to any specific domain of the ATG protein? Why did they target these exons? Please, provide some data on the characteristics of these mutants. Did they produce null alleles in all cases? They also must indicate the specific alleles used in each figure, not only a generic *atg5* or *atg7*.

Reply: The information regarding to the CRISPR targeted domain of the ATG proteins has been incorporated into Figure 2. When designing the gRNAs targeting *ATG* genes, we prioritized target specificity and editing efficiency to ensure a more effective impairment of ATG protein expression.

To further characterize these mutants, schematic diagrams depicting mutant versions of ATG proteins created by CRISPR/Cas9-edited mutagenesis have been added in the revised Figure 2. In addition, the protein levels of ATG5 in WT and three independent *atg5* mutant lines were assessed using ATG5 immunoblotting. The results revealed that ATG5 protein was barely detected in *atg5* mutants compared to WT plants. Similarly, ATG7 protein levels were also hardly detected in *atg7* mutant lines as determined by ATG7 immunoblotting. New data about these findings have been added to the revised Figure 2. Furthermore, specific alleles used in each experiment have been indicated in the revised manuscript according to your suggestion.

Q16. They should provide data on autophagosomes in *atg7* mutants, not only in *atg5*, as they use both to detect defects in seed development.

Reply: To assess autophagic activity in *atg7* mutants, we employed two independent approaches: visualization of autophagosomes using ATG8 immunofluorescence and analysis of the lipidated forms of ATG8 (ATG8-PE) through ATG8 immunoblotting. Similar to the findings in *atg5* mutants, the number of autophagosomes was significantly reduced in *atg7* mutants as observed through ATG8 immunofluorescence (Supplementary Fig. 4). Furthermore, the results from ATG8 immunoblotting confirmed these findings, showing a significant decrease in the levels of ATG8-PE in *atg7* mutants compared to WT plants. These new results have been added in revised Figure 2h.

Q17. Line 145 “exhibited delayed degradation”. Does it mean that the endothelium in *atg5* mutant is finally degraded? They must show this data or rewrite the text.

Reply: According to your comments, the development of the endothelium was examined in *atg5-2* mutants and wild-type (WT) plants at 8 DAP using pseudo-Schiff PI staining. The results revealed that the endothelium in *atg5* mutants still expanded more significantly compared to WT plants. Therefore, in order to express it more accurately, we have rewritten the text based on your suggestion. As a result, the sentence "and exhibited delayed degradation" has been removed in the revised manuscript.

Q18. Lines 165-171. Data on Figure 3E is poorly described. It is not clear whether all embryos in *atg* mutants are defective, and are classified in three groups, or some embryos have normal WT morphology, therefore mutant seeds can be classified in 4 groups, the WT (normal embryos) plus 3 categories of defective embryos. This part of the text should be rewritten to properly describe the embryo phenotype of the mutant.

Reply: Around 40% of embryos in the *atg5* mutants exhibit defects, which can be classified into three groups based on their morphology. In the revised manuscript, Figure 3E has been updated according to your suggestion to display four groups of embryos, including normal embryos and three categories of defective embryos. In addition, the description of the *atg5* embryo phenotype has been rewritten to reflect these changes in the revised manuscript.

Q19. Line 179-182. Indicate that TUNEL is used to detect fragmented DNA. Change the text as the TUNEL or TEM is not done only in integumentary tapetum cells, but rather to the whole seed (“we subjected the integumentary tapetum to...”).

Reply: This text has been revised according to your suggestion. The phrase "integumentary tapetum" has been removed from the revised manuscript.

Q20. Line 183-184. How is the TUNEL signal calculated? Please clarify what “52.2%; 47 out of 90” means.

Reply: TUNEL analysis was independently performed three times, with 30 seeds in each assay, resulting in a total of 90 seeds being analyzed. The percentage of TUNEL signal was calculated as the number of seeds showing TUNEL signals in the integumentary tapetum divided by the total number of seeds. For example, "52.2%; 47 out of 90" indicates

that TUNEL signals were observed in the integumentary tapetum of 47 out of 90 seeds, equating to 52.2%. To provide a clearer expression of the TUNEL signal data, the revised manuscript has clarified these details.

Q21. What are the two images in Figure 5A? It seems that the lower panel is the GFP signal and the upper is the merged GFP plus the mPS-PI staining. Clearly indicate that in Figure and Figure legend.

Reply: The lower panel displays the GFP signal, while the upper panel shows the merged GFP with the SR2200 staining. Descriptions of images have been added in both Figure 5A and the figure legends.

Q22. What does “Zygote” mean in Figure 5E? It seems that they refer to WT embryo/seed? Clarify and unify nomenclature with Figure 3E (where they refer to WT embryo).

Reply: Yes, you are right. It means WT embryo. The nomenclature in Figure 3E has been standardized. "Zygote" has been replaced with "normal embryo", and the term "abnormal" has been revised to "abnormal embryo" according to your suggestion.

Q23. Line 226. To me, data resented does not “indicate” the existence of cell-to-cell communication. Indeed, it seems that ATG5 is important for endothelium development which has an impact in embryo and seed development, but it does not indicate a cell-to-cell communication. The last sentence in this paragraph must be rewritten to reflect the experimental data presented. They could indicate that their data may suggest a possible communication mechanism between the integumentary tapetum and the embryo.

Reply: The sentence has been revised as per your suggestion. The word "indicate" has been changed to "suggest a possible" according to your suggestion.

Q24. Split and rewrite sentence in lines 141-144 “We observed clear developmental defects in the seed coat and the embryo in *atg5* mutants. In WT seeds, the integumentary tapetum expanded from 0 to 4 DAP, and then gradually degraded. In contrast, the integumentary tapetum of *atg5* mutant seeds ...”

Reply: The sentence in lines 141-144 has been separated and rephrased based on your suggestion.

Q25. Line 167, introduce the Figure 3E at the end of this sentence, after the word “lineages”, not several sentences later.

Reply: The introduction regarding Figure 3E has been relocated following the word “lineages” according to your suggestion.

Q26. It is quite difficult to differentiate the colors used for WT and *atg5* in Figure 7C and D. Color blind readers will appreciate the use of other colors.

Reply: The colors in Figure 7C and D have been adjusted for the benefit of color-blind readers.

Q27. The Figure legends should be unified in style and the way data is presented. For example, in most figures, scale bars are indicated in the corresponding panels but in Figure

7 it is indicated at the end of the legend, not after panels A and B.

Reply: The style of the figure legends and the method of data presentation have been revised to achieve uniformity according to your suggestion. Bar-dot plots were used to present data with a sample size of $n \leq 3$, while box-whisker plots were used to present data with a sample size of $n \geq 5$.

Q28. Reference 41 is introduced late. Move it to line 345.

Reply: Reference 41 has been moved to line 345 according to your suggestion.

Q29. Line 328. Down-emphasize “orchestrates” in the title of this section. It seems clear that integumentary tapetum is important for embryo development, but I am not sure we can say that it “orchestrates” embryo development, unless more data is provided.

Reply: The word "orchestrates" has been changed to "contributes".

Response to Reviewer #3

Q1. Zhao et al., investigated the functions of autophagy in seed development, focusing on its centrality to the timely execution of programmed cell death (PCD) within the integumentary tapetum, analogous to the anther tapetum. Upon autophagy inhibition, a delay in the PCD of the integumentary tapetum was observed. Furthermore, it resulted in aberrations in embryo patterning. The targeted re-establishment of autophagic functions indicates the non-autonomous role of the integumentary tapetum in embryo patterning. Additionally, lipidomic analysis reveals distinct stage-dependent contribution of autophagy to seed lipid metabolism. The study finally reveals that autophagy affects triacylglycerol accumulation in integumentary tapetum, highlighting its criticality for seed viability. This study offers significant insights into the implications of autophagy in the integumentary tapetum. However, I harbor several pivotal inquiries and concerns that need to be addressed. Additionally, I proffer a few recommendations that could potentially enhance the robustness of their findings. My detailed comments are delineated below:

Reply: Thanks for your comments, which have greatly helped us to improve the manuscript.

Q2. In Figure 1, the punctate dots depicted in Figure 1c are not sufficiently convincing as autophagosomes which are shown by the autophagosome reporter GFP-ATG8 shown in Figure 1d. A colocalization study using immunofluorescence labeling with the anti-ATG5 antibody in the integumentary tapetum expressing GFP-ATG8 would better support this conclusion. Additionally, Figure 1 lacks details on the subcellular localization and distribution of ATG7 in the integumentary tapetum.

Reply: We employed two independent approaches, including GFP-ATG8 and ATG8 immunofluorescence, to visualize autophagosomes, as illustrated in Figure 1c and d. Upon revising our manuscript in accordance with your recommendations, we found that GFP-ATG8 fluorescence within the integumentary tapetum was not readily detectable following a sequence of immunofluorescence procedures. This diminished fluorescence was likely attributable to quenching effects during the ATG8 immunofluorescence process. Consequently, to validate the results according to your suggestion, we employed integumentary tapetum cells expressing GFP-ATG8 to perform dual immunofluorescence assays for both ATG8 and GFP antibodies. Our observations confirm that autophagosomes labeled with GFP-ATG8 exhibit notable colocalization with those identified through ATG8 immunofluorescence staining in the integumentary tapetum. We have included an additional figure in the Supplementary Information (Supplementary Figure 3).

Furthermore, we have investigated the subcellular localization of ATG7 in the integumentary tapetum through ATG7 immunofluorescence. The new results, along with the subcellular localization of ATG5 in the integumentary tapetum, have been incorporated into a new supplementary Figure 2.

Q3. In addition, the decreased autophagy level in the CRISPR/Cas9-generated *atg5* and *atg7* mutants needs verification in comparison to the WT. Simply calculating the number of fluorescent dots in the integumentary tapetum is not sufficiently convincing. To more effectively demonstrate the impairment of autophagy, a protein-level comparison between the mutants and the WT is also necessary.

Reply: To further confirm the deficiency of autophagy in *atg* mutants, we analyzed the levels of lipidated forms of ATG8 in WT plants, *atg5* mutants, and *atg7* mutants through immunoblotting. In line with the results obtained from autogosome quantification, the level of lipidated forms of ATG8 (ATG8-PE) significantly decreased in the *atg* mutants compared to that in the WT plants (Figure 2h), providing further confirmation of impaired autophagy.

Q4. A pressing question arising from this study is: How does autophagy influence cell patterning or distribution during embryo formation? What molecular links exist between autophagy and embryo pattern formation, and what are the underlying mechanisms? While the authors emphasize this in the title and abstract, supporting data is sparse with only Figure 3e directly addressing this. As such, the conclusions would be more convincing by benefiting from more supporting experimental data.

Reply: Thanks for your comments. Based on your suggestions, we performed two additional experiments. Firstly, in order to validate the role of autophagy in embryo pattern formation, we carefully analyzed the embryo phenotype in *atg7* mutants, which are also deficient in autophagy. Consistent with the embryonic defects observed in *atg5* mutant, we observed similar embryo developmental defects in *atg7-1* mutants. These results further support the role of autophagy in embryo pattern formation. New data have been added in supplementary Figure 6d.

Secondly, to further investigate potential molecular links and underlying mechanisms between autophagy in the integumentary tapetum and embryo pattern formation, we isolated embryos from WT plants, *atg5* mutants, and *atg5* mutants expressing *proTPE8:ATG5-GFP* (This strategy is designed to selectively re-establish *ATG5* expression in the integumentary tapetum of *atg5* mutants, while ensuring that *ATG5* is not expressed in the *atg5* embryo itself.) for RNA-sequencing. Our transcriptomic analysis provides us with an opportunity to investigate molecular links between autophagy and embryo pattern formation. Comparative results revealed that approximately 3,000 genes are differentially regulated in *atg5* embryos compared to WT plants. Importantly, with specific expression of *ATG5* in the integumentary tapetum driven by the *TPE8* promoter, the number of differentially regulated genes significantly decreased in the *atg5* embryos. This reduction suggests that specific expression of *ATG5* in the integumentary tapetum can partially restore the aberrant embryonic gene expression profile observed in *atg5* mutants. We then focused on the analysis of the set of genes in embryos that are regulated by autophagy in the integumentary tapetum. The results unveiled several well-known pathways associated with embryo pattern formation, including the pattern specification- and auxin-related pathways, are regulated by autophagy in the integumentary tapetum. These insightful findings have been integrated into the new figure 6. Furthermore, to more accurately express the results about autophagy and embryo pattern, the text have been improved to avoid overstatement.

Q5. In Figure 4, the authors observed delayed PCD in the integumentary tapetum of the *atg5* mutant. Yet, the intricate relationship between integumentary tapetum PCD and autophagy remains ambiguous and warrants further evidence to solidify the conclusion. It's crucial to discern whether the delayed PCD in the integumentary tapetum directly or

indirectly stems from impaired autophagy. Additionally, the mechanism by which autophagy might instigate this delayed PCD needs elucidation.

Reply: Indeed, exploring the molecular mechanisms linking autophagy and programmed cell death (PCD) in the integumentary tapetum is an important area of investigation in the field of plant developmental biology, especially because this area is currently not well understood. In recent years, the potential relationship between autophagy and programmed cell death (PCD) has been discussed. Although the direct link between PCD and autophagy is still largely unknown, there is evidence suggesting that autophagy may facilitate PCD. For example, vacuolar processing enzyme (VPE), which exhibits caspase-1-like activity, has been identified as a crucial cysteine protease in PCD in various cell types, functioning as an executor protease in plant PCD similar to animal cell death caspase 1 (Hatsugai et al., 2004). In addition, recent research has also demonstrated that VPE translocates to the cell vacuole through the autophagy pathway, thereby resulting in PCD (Teper-Bamnolker et al., 2021). These data suggests VPE may serve as a potential link between autophagy and PCD.

Despite the challenges in elucidating the relationship between integumentary tapetum PCD and autophagy at this stage, we conducted two new experiments to explore the potential connection between autophagy and PCD in the integumentary tapetum. Firstly, through TUNEL analysis, we observed delayed PCD in the integumentary tapetum of the *atg7* mutant, further confirming the delayed PCD in autophagy-deficient mutants (Figure 4a,c). Secondly, the proteolytic activities of vacuolar processing enzyme in the WT and *atg5* ovules and seeds to investigate whether the role of autophagy in integumentary tapetum PCD is mediated through VPE. Consistent with the delayed PCD in *atg5-2* mutants, we observed a significant decrease in VPE proteolytic activities in *atg5-2* seeds, but not in the ovules. This suggests that autophagy-mediated integumentary tapetum PCD likely occurs through regulating the activities of caspase 1-like protease VPE. Understanding this potential link could provide valuable insights into the relationship between integumentary tapetum PCD and autophagy. The new data have been added to the revised manuscript (Figure 4d).

Q6. It is confusing that why the authors use the ovules and 4-DAP seeds from WT and *atg5* mutant for the lipid metabolism analysis especially since most prior observations and data relate to the integumentary tapetum. Considering seed development as a consequence of developmental defects in the integumentary tapetum, it seems more logical and convincing to analyze lipid metabolism directly within the integumentary tapetum. Doing so would provide unequivocal evidence of autophagy's role in regulating lipid metabolism in integumentary tapetum cells.

Reply: Isolating the integumentary tapetum in a seed presents technical challenges for lipidomic analysis, as lipidomic analysis typically requires a significant amount of material. It is nearly impossible to collect enough integumentary tapetum material for lipidomic analysis using current techniques. Therefore, we collected ovules and 4-DAP seeds for comprehensive lipidomic analysis. Subsequently, we investigated lipid metabolism, focusing primarily on triacylglycerol in the integumentary tapetum through BODIPY staining. This approach allowed us to demonstrate the role of autophagy in regulating lipid

metabolism in integumentary tapetum cells.

To further confirm the role of autophagy in integumentary tapetum lipid metabolism, we also compared the level of triacylglycerol in the integumentary tapetum of *atg7* mutants with that of WT plants using BODIPY staining. Similar to the results observed in *atg5* mutants, we found higher triacylglycerol content in *atg7* ovules before fertilization, but lower levels in *atg7* seeds after fertilization. These new findings have been integrated into the revised Figure 8.

Q7. The mere observation of reduced triacylglycerol accumulation in the integumentary tapetum of *atg5* mutants seems cursory. It's essential to furnish evidence detailing how autophagy might directly influence this accumulation. How can we exclude the possibility that triacylglycerol accumulation isn't directly orchestrated by autophagy but rather manifests as a secondary effect in autophagy-deficient mutants?

Reply: The relationship between autophagy and lipid metabolism is currently an interesting research area, and is waiting to be further investigated in the field of plant biology. Although the mechanisms by which autophagy regulates lipid metabolism are still largely unknown, several reports have demonstrated the accumulation of triacylglycerol (TG) in autophagy-deficient mutants (Kurusu et al., 2014; Avin-Wittenberg et al., 2015; Fan et al., 2019), suggesting the involvement of autophagy in TG degradation. In addition, several pieces of evidence also support the role of autophagy in TG synthesis (Fan et al., 2019). All these studies suggest a complex relationship between autophagy and lipid metabolism.

In response to your comments, we performed two additional experiments. Firstly, we evaluated the TG content in the integumentary tapetum of *atg7* mutants, another independent mutant with autophagy deficiency, using BODIPY staining. Consistent with the lipidomic and BODIPY staining analysis results of *atg5* mutants, we observed higher triacylglycerol contents in integumentary tapetum of *atg7-1* ovules before fertilization, but lower triacylglycerol levels in integumentary tapetum of *atg7-1* seeds after fertilization, further confirming the role of autophagy in regulating TG accumulation.

Next, we compared the expression levels of genes related to lipid biosynthesis and catabolism in WT and *atg5-2* mutants to test whether the change in TG levels in *atg5* mutants is due to the regulation of genes associated with lipid biosynthesis and catabolism. Before fertilization, we observed an increase in the expression levels of genes associated with lipid biosynthesis, while there were no significant changes in the expression levels of genes related to TG catabolism in the *atg5-2* ovules. This result is consistent with the increased TG level detected in the *atg5-2* ovules. However, in the *atg5-2* 4-DAP seeds, most of the genes associated with lipid biosynthesis and metabolism did not exhibit significant changes. This suggests that the relationship between autophagy and lipid catabolism is more intricate. Reduced TG accumulation in *atg5-2* seeds is not primarily due to change of the transcriptional levels of genes associated with TG biosynthesis or catabolism. The data about are listed in the Figure 2 below. Currently, it is difficult to distinguish whether triacylglycerol accumulation is a direct effect of autophagy or a secondary effect in autophagy-deficient mutants. The connection between autophagy and lipid metabolism in plants still requires further investigation.

Figure 2. Relative expression levels of genes associated with TG biosynthesis and catabolism in the WT and *atg5* mutants.

a,b Relative expression levels of genes associated with TG biosynthesis and catabolism in *atg5-2* ovules (**a**) and 4-DAP seeds (**b**). mRNA levels of genes associated with TG biosynthesis and catabolism in the WT were set to 1. Data represent the mean \pm SD ($n = 3$). (Two-tailed Student's *t* test, ns, no significant difference, $P > 0.05$; * $P < 0.05$; ** $P < 0.01$; *** $P < 0.001$).

Q8. The Introduction lacks comprehensive content. A concise overview of the biological roles of ATG5 and ATG7 in autophagy would be beneficial.

Reply: An introduction regarding the biological roles of ATG5 and ATG7 in autophagy has been included in the introduction section.

Q9. The letter “t” of Student’s *t* test should be in italic.

Reply: The letter “t” in “Student’s *t* test” has been changed to italic font.

Q10. Fig. 3a, the scale bars of 6 DAP are too small to be easily discerned.

Reply: The scale bars for the 6 DAP seeds in Figure 3 have been adjusted to enhance visibility and improve discernibility.

Q11. Figure 4c lacks indicated scale bar lengths. In Figure 4d, scale bar lengths are only provided in the enlarged images, not in the original ones.

Reply: The scale bar lengths for Figure 4c, as well as for both the original and enlarged images in Figure 4d, have been included in the updated figure legends.

Q12. The red signal in Figure 5 is neither labeled nor described.

Reply: The red signal is from SR2200 staining, as indicated in the updated Figure 5.

Q13. There's inconsistency in the reference style. For instance, names of mutants and species aren't italicized in the references (Lines 639, 656). They should be uniformed in reference style.

Reply: The reference style in the manuscript has been revised to ensure uniformity.

References:

- Avin-Wittenberg, T., Bajdzienko, K., Wittenberg, G., Alseekh, S., Tohge, T., Bock, R., Giavalisco, P., and Fernie, A.R. (2015). Global analysis of the role of autophagy in cellular metabolism and energy homeostasis in Arabidopsis seedlings under carbon starvation. *Plant Cell* 27, 306–322.
- Barros, J.A.S., Siqueira, J.A.B., Cavalcanti, J.H.F., Araujo, W.L., and Avin-Wittenberg, T. (2020). Multifaceted roles of plant autophagy in lipid and energy metabolism. *Trends Plant Sci.* 25, 1141–1153.
- Chang, T.K., Shrivage, B.V., Hayes, S.D., Powers, C.M., Simin, R.T., Harper, J.W., and Baehrecke, E.H. (2013). Uba1 functions in Atg7- and Atg3-independent autophagy. *Nat. Cell Biol.* 15, 1067-1078.
- Choi, H.S., Bjornson, M., Liang, J.B., Wang, J.Z., Ke, H.Y., Hur, M., De Souza, A., Kumar, K.S., Mortimer, J.C., and Dehesh, K. (2023). COG-imposed Golgi functional integrity determines the onset of dark-induced senescence. *Nat. Plants* 9, 1891–1901.
- Dall'Armi, C., Devereaux, K.A., and Di Paolo, G. (2013). The role of lipids in the control of autophagy. *Curr. Biol.* 23, R33–R45.
- Fan, J., Yu, L., and Xu, C. (2019). Dual role for autophagy in lipid metabolism in Arabidopsis. *Plant Cell* 31, 1598–1613.
- Gomez, R.E., Chambaud, C., Lupette, J., Castets, J., Pascal, S., Brocard, L., Noack, L., Jaillais, Y., Joubès, J., and Bernard, A. (2022). Phosphatidylinositol-4-phosphate controls autophagosome formation in Arabidopsis thaliana. *Nat. Commun.* 13, 4385.
- Hatsugai, N., Kuroyanagi, M., Yamada, K., Meshi, T., Tsuda, S., Kondo, M., Nishimura, M., and Hara-Nishimura, I. (2004). A plant vacuolar protease, VPE, mediates virus-induced hypersensitive cell death. *Science* 305, 855–858.

- Huang L., Wen X., Jin L., Han H.H., and H.W., G. (2024). HOOKLESS1 acetylates AUTOPHAGY-RELATED PROTEIN18a to promote autophagy during nutrient starvation in Arabidopsis. *Plant Cell* 36, 136–157.
- Kurusu, T., Koyano, T., Hanamata, S., Kubo, T., Noguchi, Y., Yagi, C., Nagata, N., Yamamoto, T., Ohnishi, T., Okazaki, Y., Kitahata, N., Ando, D., Ishikawa, M., Wada, S., Miyao, A., Hirochika, H., Shimada, H., Makino, A., Saito, K., Ishida, H., Kinoshita, T., Kurata, N., and Kuchitsu, K. (2014). OsATG7 is required for autophagy-dependent lipid metabolism in rice postmeiotic anther development. *Autophagy* 10, 878-888.
- Singh, R., Kaushik, S., Wang, Y., Xiang, Y., Novak, I., Komatsu, M., Tanaka, K., Cuervo, A.M., and Czaja, M.J. (2009). Autophagy regulates lipid metabolism. *Nature* 458, 1131–1135.
- Teper-Bamnolker, P., Danieli, R., Peled-Zehavi, H., Belausov, E., Abu-Abied, M., Avin-Wittenberg, T., Sadot, E., and Eshel, D. (2021). Vacuolar processing enzyme translocates to the vacuole through the autophagy pathway to induce programmed cell death. *Autophagy* 17, 3109–3123.
- Zhao, J., Xin, H.P., Qu, L.H., Ning, J., Peng, X.B., Yan, T.T., Ma, L.G., Li, S.S., and Sun, M.X. (2011). Dynamic changes of transcript profiles after fertilization are associated with transcription and maternal elimination in tobacco zygote, and mark the onset of the maternal-to-zygotic transition. *Plant J.* 65, 131–145.
- Zhao, P., Zhou, X.M., Zhang, L.Y., Wang, W., Ma, L.G., Yang, L.B., Peng, X.B., Bozhkov, P.V., and Sun, M.X. (2013). A bipartite molecular module controls cell death activation in the basal cell lineage of plant embryos. *PLoS Biol* 11, e1001655.

Reviewer #1 (Remarks to the Author):

The authors answered my question very well and suggested its publication.

Reviewer #2 (Remarks to the Author):

After thoroughly reviewing the revised manuscript, I concur with the implemented changes, which effectively address my previous recommendations made for the initial version of this work. Consequently, I have no additional modifications to propose.

Reviewer #3 (Remarks to the Author):

The authors have fully addressed my concerns and questions. I believe the manuscript is now suitable for publication in Nature Communications.

Reviewer #1 (Remarks to the Author):

Q1. The authors answered my question very well and suggested its publication.

Reply: Thank you for your appreciation of our revised manuscript.

Reviewer #2 (Remarks to the Author):

Q2. After thoroughly reviewing the revised manuscript, I concur with the implemented changes, which effectively address my previous recommendations made for the initial version of this work. Consequently, I have no additional modifications to propose.

Reply: Thank you for your comments and appreciation of our revised manuscript.

Reviewer #3 (Remarks to the Author):

Q3.The authors have fully addressed my concerns and questions. I believe the manuscript is now suitable for publication in Nature Communications.

Reply: Thank you for your appreciation of our revised manuscript.